# OTOv3: Towards Automatic Sub-Network Search Within General Super Deep Neural Networks

## Abstract

Existing neural architecture search (NAS) methods typically rely on pre-specified super deep neural networks (super-networks) with handcrafted search spaces beforehand. Such requirements make it challenging to extend them onto general scenarios without significant human expertise and manual intervention. To overcome the limitations, we propose the third generation of Only-Train-Once (OTOv3). OTOv3 is perhaps the first automated system that trains general super-networks and produces high-performing sub-networks in the one shot manner without pretraining and fine-tuning. Technologically, OTOv3 delivers three noticeable contributions to minimize human efforts: *(i)* automatic search space construction for general super-networks; *(ii)* a Hierarchical Half-Space Projected Gradient (H2SPG) that leverages the dependency graph to ensure the network validity during optimization and reliably produces a solution with both high performance and hierarchical group sparsity; and *(iii)* automatic sub-network construction based on the super-network and the H2SPG solution. Numerically, we demonstrate the effectiveness of OTOv3 on a variety of super-networks, including StackedUnets, SuperResNet, and DARTS, over benchmark datasets such as CIFAR10, Fashion-MNIST, ImageNet, STL-10, and SVNH. The sub-networks computed by OTOv3 achieve competitive even superior performance compared to the super-networks and other state-of-the-arts.

## 1 Introduction

Deep neural networks (DNNs) have achieved remarkable success in various fields, which success is highly dependent on their sophisticated underlying architectures (LeCun et al., 2015; Goodfellow et al., 2016). To design effective DNN architectures, human expertise have handcrafted numerous popular DNNs such as ResNet (He et al., 2016) and transformer (Vaswani et al., 2017). However, such human efforts may not be scalable enough to meet the increasing demands for customizing DNNs for diverse tasks. To address this issue, Neural Architecture Search (NAS) has emerged to automate the network creations and reduce the need for human expertise (Elsken et al., 2018).

Among current NAS studies, gradient-based methods (Liu et al., 2018; Yang et al., 2020; Xu et al., 2019; Chen et al., 2021b) are perhaps the most popular because of their efficiency. Such methods build an over-parameterized super-network covering all candidate connections and operations, parameterize operations via introducing auxiliary architecture variables with weight sharing, then search a (sub)optimal sub-network via formulating and solving a multi-level optimization problem.

Despite the advancements in gradient-based methods, their usage is still limited due to certain inconvenience. In particular, their automation relies on manually determining the search space for a pre-specified super-network beforehand, and requires the manual introduction of auxiliary architecture variables onto the prescribed search space. To extend these methods onto other super-networks, the users still need to manually construct the search pool, then incorporate the auxiliary architecture

variables along with building the whole complicated multi-level optimization training pipeline. The whole process necessitates significant domain-knowledge and engineering efforts, thereby being inconvenient and time-consuming for users. Therefore, it is natural to ask whether we could reach an

**_Objective._** _Given a general super-network, automatically generate its search space, train it once, and construct a sub-network that achieves a dramatically compact architecture and high performance._

Achieving the objective is severely challenging in terms of both engineering developments and algorithmic designs, consequently not achieved yet by the existing NAS works to the best of our knowledge. However, the objective has been recently achieved in an analogous task so-called structured pruning (Lin et al., 2019) by the second generation of Only-Train-Once framework (OTOv2) (Chen et al., 2021a, 2023). From

| | OTOv3 | OTOv2 | Other NAS |
|---|---|---|---|
| **General DNNs** | ✓ | ✓ | ✗ |
| **Autonomy** | ✓ | ✓ | ✓✗ |
| **Remove Connections** | ✓ | ✗ | ✓ |
| **Remove Operations** | ✓ | ✗ | ✓ |
| **Slim Operations** | ✓† | ✓ | ✗ |

† Support while is not the focus and discussed in this work.

the perspective of computational graph, the standard NAS could be considered as removing entire redundant connections (cutting edges) and operations (vertices) from super-networks. Structured pruning can be largely interpreted as a complementary NAS that removes the redundancy inside each vertex (slims operations) but preserves all the connections. OTOv2 first achieves the objective in the view of structured pruning that given a general DNN, automatically trains it only once to achieve both high performance and a slimmer model architecture without pre-training and fine-tuning.

We now build the third-generation of Only-Train-Once (OTOv3) that reaches the objective from the perspective of the standard NAS. OTOv3 automatically generates a search space given a general super-network, trains and identifies redundant connections and vertices, then builds a sub-network that achieves both high performance and compactness. As the library usage presented aside, the

```
OTOv3 Library Usage
1  from only_train_once import OTO
2  # General Super-Network
3  oto = OTO(super_net, cut_edges=True)
4  optimizer = oto.h2spg()
5  # Train as normal
6  optimizer.step()
7  oto.construct_subnet(cut_edges=True)
```

whole procedure can be automatically proceeded, dramatically reduce the human efforts, and fit for general super-networks and applications. Our main contributions can be summarized as follows.

- **Infrastructure for Automated General Super-Network Training and Sub-Network Searching.** We propose OTOv3 that perhaps the first automatically trains and searches within a general super-network to deliver a compact sub-network by erasing redundant connections and operations in the one-shot manner. As the previous OTO versions, OTOv3 trains the super-network only once without the need of pre-training and fine-tuning and is pluggable into various deep learning applications.

- **Automated Search Space Generation.** We propose a novel graph algorithm to automatically explore and establish a dependency graph given a general super-network, then analyze the dependency to form a search space consisting of minimal removal structures. The corresponding trainable variables are then partitioned into so-called generalized zero-invariant groups (GeZIGs).

- **Hierarchical Half-Space Projected Gradient (H2SPG).** We propose a novel H2SPG optimizer that perhaps the first solves a hierarchical structured sparsity problem for general DNNs. H2SPG computes a solution $x_{\text{H2SPG}}^*$ of both high performance and desired hierarchical group sparsity in the manner of GeZIGs. Compared to other optimizers, H2SPG considers the hierarchy of dependency graph to produce sparsity for ensuring the validness of the subsequent sub-network.

- **Automated Sub-Network Construction.** We propose a novel graph algorithm to automatically construct a sub-network upon the super-network parameterized as $x_{\text{H2SPG}}^*$. The resulting sub-network returns the exact same outputs as the super-network thereby no need of further fine-tuning.

- **Experimental Results.** We demonstrate the effectiveness of OTOv3 on extensive super-networks including StackedUnets, SuperResNet and DARTS, over benchmark datasets including CIFAR10, Fashion-MNIST, ImageNet, STL-10, and SVNH. OTOv3 is the first framework that could automatically deliver compact sub-networks upon general super-networks to the best of our knowledge. Meanwhile the sub-networks exhibit competitive even superior performance to the super-networks.

## 2   Related Work

**Neural Architecture Search (NAS).**   Early NAS works utilized reinforcement learning and evolution techniques to search for high-quality architectures (Zoph & Le, 2016; Pham et al., 2018; Zoph et al., 2018), while they were computationally expensive. Later on, differentiable (gradient-based)

methods were introduced to accelerate the search process. These methods start with a super-network covering all possible connection and operation candidates, and parameterize them with auxiliary architecture variables. They establish a multi-level optimization problem that alternatingly updates the architecture and network variables until convergence (Liu et al., 2018; Chen et al., 2019; Xu et al., 2019; Yang et al., 2020; Hosseini & Xie, 2022). However, these methods require a significant amount of **handcraftness** from users in advance to **manually** establish the search space, introduce additional architecture variables, and build the multi-level training pipeline. The sub-network construction is also network-specific and not flexible. All requirements necessitate remarkable domain-knowledge and expertise, making it difficult to extend to general super-networks and broader scenarios.

**Automated Structured Pruning for General DNNs.** Structure pruning is an orthogonal but related paradigm to standard NAS. Rather than removing entire operations and connections, it focuses on slimming individual vertices (Han et al., 2015). Similarly, prior structure pruning methods also required numerous handcraftness and domain knowledge, which limited their broader applicability. However, recent methods such as OTOv2 (Chen et al., 2023) and DepGraph (Fang et al., 2023) have made progress in automating the structure pruning process for general DNNs. OTOv2 is a one-shot method that does not require pre-training or fine-tuning, while DepGraph involves a multi-stage training pipeline that requires some manual intervention. In this work, we propose the third-generation version of OTO that enables automatic sub-network searching and training for general super-networks.

**Hierarchical Structured Sparsity Optimization.** We formulate the underlying optimization problem of OTOv3 as a hierarchical structured sparsity problem. Its solution possesses high group sparsity indicating redundant structures and obeys specified hierarchy. There exist deterministic optimizers solving such problems via introducing latent variables (Zhao et al., 2009), while are impractical for stochastic DNN tasks. Meanwhile, stochastic optimizers rarely study such problem. In fact, popular stochastic sparse optimizers such as HSPG (Chen et al., 2021a), DHSPG (Chen et al., 2023), proximal methods (Xiao & Zhang, 2014) and ADMM (Lin et al., 2019) overlook the hierarchy constraint. Incorporating them into OTOv3 typically delivers invalid sub-networks. Therefore, we propose H2SPG that considers graph dependency to solve it for general DNNs.

# 3 OTOv3

OTOv3 is an automated one-shot system that trains a general super-network and constructs a sub-network. The produced sub-network is not only high-performing but also has a dramatically compact architecture that is suitable for various shipping environments. The entire process minimizes the need for human efforts and is suitable for general DNNs. As outlined in Algorithm 1, given a general super-network $\mathcal{M}$, OTOv3 first explores and establishes a dependency graph. Upon the dependency graph, a search space is automatically constructed and corresponding trainable variables are partitioned into generalized zero-invariant groups (GeZIGs) (Section 3.1). A hierarchical structured sparsity optimization problem is then formulated and solved by a novel Hierarchical Half-Space Projected Gradient (H2SPG) (Section 3.2). H2SPG considers the hierarchy inside the dependency graph and computes a solution $x^*_{\text{H2SPG}}$ of both high-performance and desired hierarchical group sparsity over GeZIGs. A compact sub-network $\mathcal{M}^*$ is finally constructed via removing the structures corresponding to the identified redundant GeZIGs and their dependent structures (Section 3.3). $\mathcal{M}^*$ returns the exact same output as the super-network parameterized as $x^*_{\text{H2SPG}}$, eliminating the need of fine-tuning.

---
**Algorithm 1** Outline of OTOv3.

---
1: **Input:** A general DNN $\mathcal{M}$ as super-network to be trained and searched (no need to be pretrained).
2: **Automated Search Space Construction.** Establish dependency graph and partition the trainable parameters of $\mathcal{M}$ into generalized zero-invariant groups $\mathcal{G}_{\text{GeZIG}}$ and the complementary $\mathcal{G}^C_{\text{GeZIG}}$.
3: **Train by H2SPG.** Seek a high-performing solution $x^*_{\text{H2SPG}}$ with hierarchical group sparsity.
4: **Automated Sub-Network $\mathcal{M}^*$ Construction.** Construct a sub-network upon $x^*_{\text{H2SPG}}$.
5: **Output:** Constructed sub-network $\mathcal{M}^*$ (no need to be fine-tuned).

---

## 3.1 Automated Search Space Construction

The foremost step is to automatically construct the search space for a general super-network. However, this process presents significant challenges in terms of both engineering developments and algorithmic designs due to the complexity of DNN architecture and the lack of sufficient public APIs. To overcome

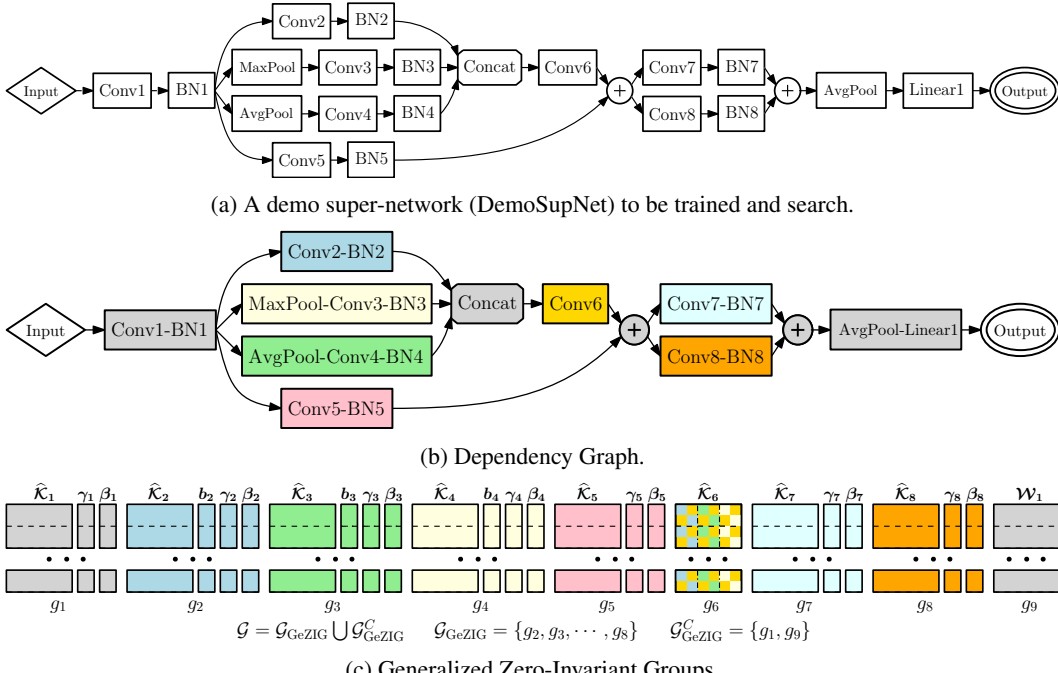

(a) A demo super-network (DemoSupNet) to be trained and search.

(b) Dependency Graph.

(c) Generalized Zero-Invariant Groups.

Figure 1: Automated Search Space Construction. $\widehat{\mathcal{K}}_i$ and $\boldsymbol{b}_i$ are the flatten filter matrix and bias vector for `Conv-i`, respectively. $\boldsymbol{\gamma}_i$ and $\boldsymbol{\beta}_i$ are the weight and bias vectors for `BN-i`. $\boldsymbol{\mathcal{W}}_i$ is the weight matrix for `Linear-i`. The columns of $\widehat{\mathcal{K}}_6$ are marked in accordance to its incoming segments.

these challenges, we propose a concept called generalized zero-invariant group (GeZIG) and formulate the search space construction as the GeZIG partition. We have also developed a dedicated graph algorithm to automatically conduct the GeZIG partition for general super-networks.

**Generalized Zero-Invariant Group (GeZIG).** The key of search space construction is to figure out the structures that can be removed from the super-network. Because of diverse roles of operations and their complicated connections inside a DNN, removing an arbitrary structure may cause the remaining DNN invalid. We say a structure *removal* if and only if the DNN after removing it is still valid. A removal structure is further said *minimal* if and only if it does not contain multiple removal structures. Zero-Invariant Group (ZIG) is proposed in (Chen et al., 2021a, 2023) that describes a class of minimal removal structures satisfying a *zero-invariant property*, *i.e.*, if all variables in ZIG equal to zero, then no matter what the input is, the output is always as zero. ZIG depicts the minimal removal structure ***inside each operation*** and is the key for realizing automatic one-shot structured pruning. We generalize ZIG as GeZIG that describes a class of minimal removal structures satisfying the zero-invariant property but ***consists of entire operations***. More illustrations regarding ZIG versus GeZIG are present in Appendix. For simplicity, throughout the paper, the minimal removal structure is referred to the counterpart consisting of operations in entirety. Consequently, automated search space construction becomes how to automatically explore the GeZIG partition for general DNNs.

**Automated GeZIG Partition.** As specified in Algorithm 2, automated GeZIG partition involves two main stages. The first stage explores the super-network $\mathcal{M}$ and establishes a dependency graph $(\mathcal{V}_d, \mathcal{E}_d)$. The second stage leverages the affiliations inside the dependency graph to find out minimal removal structures, then partitions their trainable variables to form GeZIGs. For intuitive illustrations, we elaborate the algorithm through a small but complex demo super-network depicted in Figure 1a.

**Dependency Graph Construction.** Given a super-network $\mathcal{M}$, we first construct its trace graph $(\mathcal{V}, \mathcal{E})$ displayed as Figure 1a (line 3 in Algorithm 2), where $\mathcal{V}$ represents the set of vertices (operations) and $\mathcal{E}$ represents the connections among them. As OTOv2 (Chen et al., 2023), we categorize the vertices into stem vertices, joint vertices, accessory vertices, and unknown vertices. Stem vertices refer to the operations that contain trainable variables and can transform the input tensors into different shapes, *e.g.*, `Conv` and `Linear`. The accessory vertices are the operations that may not have trainable variables and have an single input, *e.g.*, `BN` and `ReLU`. Joint vertices aggregate multiple inputs into a single output, *e.g.*, `Add` and `Concat`. The remaining vertices are considered as `unknown`.

---

**Algorithm 2** Automated Search Space Construction.

---

1: **Input:** A super-network $\mathcal{M}$ to be trained and searched.
2: ***Dependency graph construction.***
3: Construct the trace graph $(\mathcal{E}, \mathcal{V})$ of $\mathcal{M}$.
4: Initialize an empty graph $(\mathcal{V}_d, \mathcal{E}_d)$.
5: Initialize queue $\mathcal{Q} \leftarrow \{\mathcal{S}(v) : v \in \mathcal{V} \text{ is adjacent to the input of trace graph}\}$.
6: **while** $\mathcal{Q} \neq \emptyset$ **do**
7:     Dequeue the head segment $\mathcal{S}$ from $\mathcal{Q}$.
8:     Grow $\mathcal{S}$ in the depth-first manner till meet either joint vertex or multi-outgoing vertex $\hat{v}$.
9:     Add segments into $\mathcal{V}_d$ and connections into $\mathcal{E}_d$.
10:     Enqueue new segments into the tail of $\mathcal{Q}$ if $\hat{v}$ has outgoing vertices.
11: ***Find minimal removal structures.***
12: Get the incoming vertices $\widehat{\mathcal{V}}$ for joint vertices in the $(\mathcal{V}_d, \mathcal{E}_d)$.
13: Group the trainable variables in the vertex $v \in \widehat{\mathcal{V}}$ as $g_v$.
14: Form $\mathcal{G}_{\text{GeZIG}}$ as the union of the above groups, *i.e.*, $\mathcal{G}_{\text{GeZIG}} \leftarrow \{g_v : v \in \widehat{\mathcal{V}}\}$.
15: Form $\mathcal{G}_{\text{GeZIG}}^C$ as the union of the trainable variables in the remaining vertices.
16: **Return** trainable variable partition $\mathcal{G} = \mathcal{G}_{\text{GeZIG}} \cup \mathcal{G}_{\text{GeZIG}}^C$ and dynamic dependency graph $(\mathcal{V}_d, \mathcal{E}_d)$.

---

We begin by analyzing the trace graph $(\mathcal{V}, \mathcal{E})$ to create a dependency graph $(\mathcal{V}_d, \mathcal{E}_d)$, wherein each vertex in $\mathcal{V}_d$ serves as a potential minimal removal structure candidate. To proceed, we use a queue container $\mathcal{Q}$ to track the candidates (line 5 of Algorithm 2). The initial elements of this queue are the vertices that are directly adjacent to the input of $\mathcal{M}$, such as Conv1. We then traverse the graph in the breadth-first manner, iteratively growing each element (segment) $\mathcal{S}$ in the queue until a valid minimal removal structure candidate is formed. The growth of each candidate follows the depth-first search to recursively expand $\mathcal{S}$ until the current vertices are considered as endpoints. The endpoint vertex is determined by whether it is a joint vertex or has multiple outgoing vertices, as indicated in line 8 of Algorithm 2. Intuitively, a joint vertex has multiple inputs, which means that the DNN may be still valid after removing the current segment. This suggests that the current segment may be removable. On the other hand, a vertex with multiple outgoing neighbors implies that removing the current segment may cause some of its children to miss the input tensor. For instance, removing Conv1-BN1 would cause Conv2, MaxPool and AvgPool to become invalid due to the absence of input in Figure 1a. Therefore, it is risky to remove such candidates. Once the segment $\mathcal{S}$ has been grown, new candidates are initialized as the outgoing vertices of the endpoint and added into the container $\mathcal{Q}$ (line 10 in Algorithm 2). Such procedure is repeated until the end of graph traversal. Ultimately, a dependency graph $(\mathcal{V}_d, \mathcal{E}_d)$ is created, as illustrated in Figure 1b.

**Form GeZIGs.** We proceed to identify the minimal removal structures in $(\mathcal{V}_d, \mathcal{E}_d)$ to create the GeZIG partition. The qualified instances are the vertices in $\mathcal{V}_d$ that have trainable variables and all of their outgoing vertices are joint vertices. This is because a joint vertex has multiple inputs and remains valid even after removing some of its incoming structures, as indicated in line 12 in Algorithm 2. Consequently, their trainable variables are grouped together into GeZIGs (line 13-14 in Algorithm 2 and Figure 1c). The remaining vertices are considered as either unremovable or belonging to a large removal structure, which trainable variables are grouped into the $\mathcal{G}_{\text{GeZIG}}^C$ (the complementary to $\mathcal{G}_{\text{GeZIG}}$). As a result, for the super-network $\mathcal{M}$, all its trainable variables are encompassed by the union $\mathcal{G} = \mathcal{G}_{\text{GeZIG}} \cup \mathcal{G}_{\text{GeZIG}}^C$, and the corresponding structures in $\mathcal{G}_{\text{GeZIG}}$ constitute its search space.

### 3.2 Hierarchical Half-Space Projected Gradient (H2SPG)

Given a super-network $\mathcal{M}$ and its group partition $\mathcal{G} = \mathcal{G}_{\text{GeZIG}} \cup \mathcal{G}_{\text{GeZIG}}^C$, the next is to jointly search for a valid sub-network $\mathcal{M}^*$ that exhibits the most significant performance and train it to high performance. Searching a sub-network is equivalent to identifying the redundant structures in $\mathcal{G}_{\text{GeZIG}}$ to be further removed and ensures the remaining network still valid. Training the sub-network becomes optimizing over the remaining groups in $\mathcal{G}$ to achieve high performance. We formulate a hierarchical structured sparsity problem to accomplish both tasks simultaneously as follows.

$$\underset{\boldsymbol{x} \in \mathbb{R}^n}{\text{minimize}} \, f(\boldsymbol{x}), \quad \text{s.t. Cardinality}(\mathcal{G}^0) = K, \text{ and } (\mathcal{V}_d/\mathcal{V}_{\mathcal{G}^0}, \mathcal{E}_d/\mathcal{E}_{\mathcal{G}^0}) \text{ is valid,} \tag{1}$$

where $f$ is the prescribed loss function, $\mathcal{G}^{=0} := \{g \in \mathcal{G}_{\text{GeZIG}} | [\boldsymbol{x}]_g = 0\}$ is the set of zero groups in $\mathcal{G}_{\text{GeZIG}}$, which cardinality measures its size. $K$ is the target group sparsity, indicating the number of

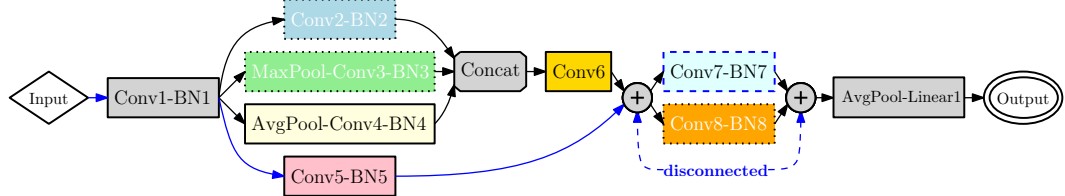

Figure 2: Check validness of redundant candidates. Target group sparsity $K = 3$. `Conv7-BN7` has larger redundancy score than `Conv2-BN2`. Dotted vertices are marked as redundant candidates.

GeZIGs that should be identified as redundant. The redundant GeZIGs are projected onto zero, while the important groups are preserved as non-zero and optimized for high performance. A larger $K$ dictates a higher sparsity level that produces a more compact sub-network with fewer FLOPs and parameters. $(\mathcal{V}_d/\mathcal{V}_{\mathcal{G}^0}, \mathcal{E}_d/\mathcal{E}_{\mathcal{G}^0})$ refers to the graph removing vertices and edges corresponding to zero groups $\mathcal{G}^0$. This graph being valid is specified for NAS that requires the zero groups distributed obeying the hierarchy of super-network to ensure the resulting sub-network functions correctly.

Problem (1) is difficult to solve due to the non-differential and non-convex sparsity constraint and the graph validity constraint. Existing optimizers such as DHSPG (Chen et al., 2023) overlook the architecture evolution and hierarchy during the sparsity exploration, which is crucial to (1). In fact, they are mainly applied for pruning tasks, where the connections and operations are preserved (but become slimmer). Consequently, employing them onto (1) usually produces invalid sub-networks.

**Outline of H2SPG.** To effectively solve problem (1), we propose a novel H2SPG to consider the hierarchy and ensure the validness of graph architecture after removing redundant vertices and connections during the optimization process. To the best of our knowledge, H2SPG is the first the optimizer that successfully solves such hierarchical structured sparsity problem (1), which outline is stated in Algorithm 3.

H2SPG is built upon the DHSPG in OTOv2 but with dedicated designs regarding the hierarchical constraint. In general, H2SPG is a hybrid multi-phase optimizer that first partitions the groups of variables into important and potentially redundant segments, then employs specified updating mechanisms onto different segments to achieve a solution with both desired hierarchical group sparsity

---

**Algorithm 3** Hierarchical Half-Space Projected Gradient

1: **Input:** initial variable $\boldsymbol{x}_0 \in \mathbb{R}^n$, initial learning rate $\alpha_0$, warm-up steps $T_w$, target group sparsity $K$, momentum $\omega$, dependency graph $(\mathcal{V}_d, \mathcal{E}_d)$ and group partitions $\mathcal{G}$.
2: *Warm-up Phase.*
3: **for** $t = 0, 1, \cdots, T_w - 1$ **do**
4:      Calculate gradient estimate $\nabla f(\boldsymbol{x}_t)$ or its variant.
5:      Update next iterate $\boldsymbol{x}_{t+1} \leftarrow \boldsymbol{x}_t - \alpha_t \nabla f(\boldsymbol{x}_t)$.
6:      Calculate redundancy score $s_{t,g}$ for $g \in \mathcal{G}_{\text{GeZIG}}$.
7:      Update $s_g \leftarrow \omega s_g + (1 - \omega) s_{t,g}$ for $g \in \mathcal{G}_{\text{GeZIG}}$.
8: Construct $\mathcal{G}_r$ and $\mathcal{G}_r^C$ given scores, $\mathcal{G}$, $(\mathcal{V}_d, \mathcal{E}_d)$, and $K$.
9: *Hybrid Training Phase.*
10: **for** $t = T_w, T_w + 1, \cdots,$ **do**
11:      Compute gradient estimate $\nabla f(\boldsymbol{x}_t)$ or its variant.
12:      Update $[\boldsymbol{x}_{t+1}]_{\mathcal{G}_r^C}$ as $[\boldsymbol{x}_t - \alpha_t \nabla f(\boldsymbol{x}_t)]_{\mathcal{G}_r^C}$.
13:      Select proper $\lambda_g$ for each $g \in \mathcal{G}_r$.
14:      Compute $[\tilde{\boldsymbol{x}}_{t+1}]_{\mathcal{G}_r}$ via subgradient descent of $\psi$.
15:      Perform Half-Space projection over $[\tilde{\boldsymbol{x}}_{t+1}]_{\mathcal{G}_r}$.
16:      Update $[\boldsymbol{x}_{t+1}]_{\mathcal{G}_r} \leftarrow [\tilde{\boldsymbol{x}}_{t+1}]_{\mathcal{G}_r}$.
17: **Return** the final iterate $\boldsymbol{x}^*_{\text{DHSPG+}}$.

---

and high performance. The variable partition considers the hierarchy of dependency graph $(\mathcal{V}_d, \mathcal{E}_d)$ to ensure the validness of the resulting sub-network graph. Vanilla stochastic gradient descent (SGD) or its variant such as Adam (Kingma & Ba, 2014) optimizes the important variables to achieve the high performance. Half-space gradient descent (Chen et al., 2021a) identifies redundant groups among the candidates and projects them onto zero without sacrificing the objective function to the largest extent.

**Warm-Up Phase.** To proceed, H2SPG first warms up all variables by conducting SGD or its variants $T_w$ steps (line 4-5 in Algorithm 3). During each warm-up step $t$, a redundancy score of each group $g \in \mathcal{G}_{\text{GeZIG}}$ is computed upon the current iterate $\boldsymbol{x}_t$ and exponentially averaged by a momentum coefficient $\omega$ (line 6-7 in Algorithm 3). Larger redundancy score indicates the group exhibits less prediction power, thus may be redundant. The redundancy score calculation is modular, where we follow DHSPG to consider the cosine similarity between negative gradient $-[\nabla f(\boldsymbol{x}_t)]_g$ and the projection direction $-[\boldsymbol{x}]_g$ as well as the average variable magnitude. After warm-up, the redundancy scores of all groups in $\mathcal{G}_{\text{GeZIG}}$ are sorted. We then perform a sanity check and select the groups with top-K redundancy scores as the redundant group candidates $\mathcal{G}_r \subseteq \mathcal{G}_{\text{GeZIG}}$. The complementary groups

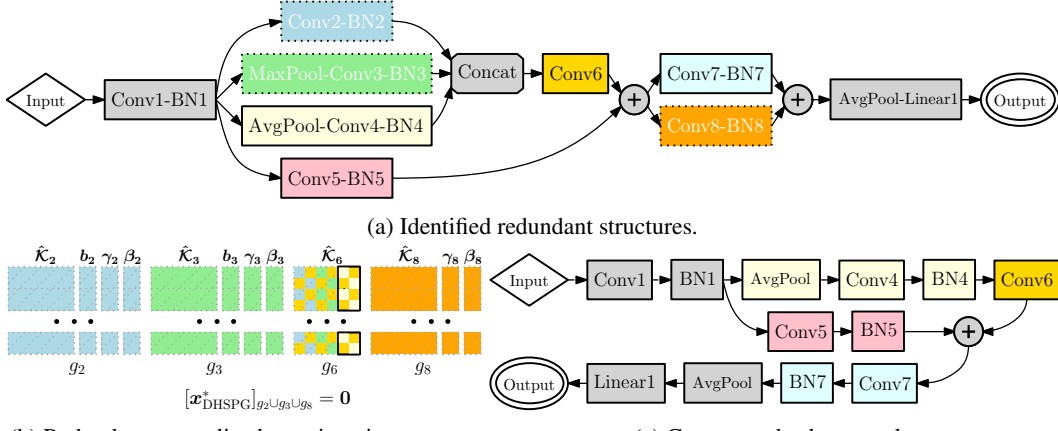

(a) Identified redundant structures.

(b) Redundant generalized zero-invariant groups.

(c) Constructed sub-network.

Figure 3: Redundant removal structures idenfitications and sub-network construction.

with lower redundancy scores are marked as important ones and form $\mathcal{G}_r^C := \mathcal{G}/\mathcal{G}_r$. The sanity check verifies whether the remaining graph is still connected after removing a vertex. If so, the current vertex is added into $\mathcal{G}_r$; otherwise, the subsequent vertex is turned into considerations. As illustrated in Figure 2, though `Conv7-BN7` has a larger redundancy score than `Conv2-BN2`, `Conv2-BN2` is marked as potentially redundant but not `Conv7-BN7` since there is no path connecting the input and the output of the graph after removing `Conv7-BN7`. This mechanism largely guarantees that even if all redundant candidates are erased, the resulting sub-network is still functioning as normal.

**Hybrid Training Phase.** H2SPG then engages into the hybrid training phase to produce desired group sparsity over $\mathcal{G}_r$ and optimize over $\mathcal{G}_r^C$ for pursuing excellent performance till the convergence. This phase mainly follows DHSPG (Chen et al., 2023), and we briefly describe the steps for completeness. In general, for the important groups of variables in $\mathcal{G}_r^C$, the vanilla SGD or its variant is employed to minimize the objective function to the largest extent (line 11-12 in Algorithm 3). For redundant group candidates in $\mathcal{G}_r$, we formulate a relaxed non-constrained subproblem as (2) to gradually reduce the magnitudes without deteriorating the objective and project groups onto zeros only if the projection serves as a descent direction for the objective during the training process (line 13-16 in Algorithm 3).

$$\underset{[\boldsymbol{x}]_{\mathcal{G}_r}}{\text{minimize}} \, \psi([\boldsymbol{x}]_{\mathcal{G}_r}) := f\left([\boldsymbol{x}]_{\mathcal{G}_r}\right) + \sum_{g \in \mathcal{G}_r} \lambda_g \left\| [\boldsymbol{x}]_g \right\|_2, \tag{2}$$

where $\lambda_g$ is a group-specific regularization coefficient and dedicately selected as DHSPG. H2SPG then performs a subgradient descent of $\psi$ over $[\boldsymbol{x}]_{\mathcal{G}_r}$, followed by a Half-Space projection (Chen et al., 2021a) to effectively produce group sparsity with the minimal sacrifice of the objective function. At the end, a high-performing solution $\boldsymbol{x}^*_{\text{H2SPG}}$ with desired hierarchical group sparsity is returned.

### 3.3 Automated Sub-Network Construction.

We finally construct a sub-network $\mathcal{M}^*$ upon the super-network $\mathcal{M}$ and the solution $\boldsymbol{x}^*_{\text{H2SPG}}$ by H2SPG. The solution $\boldsymbol{x}^*_{\text{H2SPG}}$ should attain desired target hierarchical group sparsity level and achieve high performance. As illustrated in Figure 3, we first traverse the graph to remove the entire vertices and the related edges from $\mathcal{M}$ corresponding to the redundant GeZIGs being zero, *e.g.*, `Conv2-BN2`, `MaxPool-Conv3-BN3` and `Conv8-BN8` are removed due to $[\boldsymbol{x}^*_{\text{H2SPG}}]_{g_2 \cup g_3 \cup g_8} = \boldsymbol{0}$. Then, we traverse the graph in the second pass to remove the affiliated structures that are dependent on the removed vertices to keep the remaining operations valid, *e.g.*, the first and second columns in $\hat{\mathcal{K}}_6$ are erased since its incoming vertices `Conv2-BN2` and `MaxPool-Conv3-BN3` has been removed (see Figure 3b). Next, we recursively erase unnecessary vertices and isolated vertices. Isolated vertices refer to the vertices that have neither incoming nor outgoing vertices. Unnecessary vertices refer to the skippable operations, *e.g.*, `Concat` and `Add` (between `Conv7` and `AvgPool`) become unnecessary. Ultimately, a compact sub-network $\mathcal{M}^*$ is constructed as shown in Figure 3c. By the definition of GeZIGs, the redundant GeZIGs (have been projected onto zeros) contribute none to the model outputs. Consequently, the $\mathcal{M}^*$ returns the exact same output as the super-network $\mathcal{M}$ with $\boldsymbol{x}^*_{\text{H2SPG}}$, which avoids the necessity of further fine-tuning the sub-network.[1]

---

[1]Remark here that the sub-network is still compatible to be fine-tuned afterwards if needed.

## 4 Numerical Experiments

In this section, we employ OTOv3 to one-shot automatically train and search within general super-networks to construct compact sub-networks with high performance. The numerical demonstrations cover extensive super-networks including DemoSupNet shown in Section 3, StackedUnets (Ronneberger et al., 2015; Chen et al., 2023), SuperResNet (He et al., 2016; Lin et al., 2021), and DARTS (Liu et al., 2018), and benchmark datasets, including CIFAR10 (Krizhevsky & Hinton, 2009), Fashion-MNIST (Xiao et al., 2017), ImageNet (Deng et al., 2009), STL-10 (Coates et al., 2011) and SVNH (Netzer et al., 2011). More implementation details of experiments and OTOv3 library and limitations are provided in Appendix A. The dependency graphs and the constructed sub-networks are depicted in Appendix C. Ablation studies regarding H2SPG is present in Appendix D.

Table 1: OTOv3 on extensive super-networks and datasets.

| Backend | Dataset | Method | FLOPs (M) | # of Params (M) | Top-1 Acc. (%) |
|---------|---------|--------|-----------|-----------------|----------------|
| DemoSupNet | Fashion-MNIST | Baseline | 209 | 0.82 | 84.9 |
| DemoSupNet | Fashion-MNIST | **OTOv3** | **107** | **0.45** | **84.7** |
| StackedUnets | SVNH | Baseline | 184 | 0.80 | 95.3 |
| StackedUnets | SVNH | **OTOv3** | **115** | **0.37** | **96.1** |
| DARTS (8 cells) | STL-10 | Baseline | 614 | 4.05 | 74.6 |
| DARTS (8 cells) | STL-10 | **OTOv3** | **127** | **0.64** | **75.1** |

**DemoSupNet on Fashion-MNIST.** We first experiment with the DemoSupNet presented as Figure 1a on Fashion-MNIST. OTOv3 automatically establishes a search space of DemoSupNet and partitions its trainable variables into GeZIGs. H2SPG then trains DemoSupNet from scratch and computes a solution of high performance and hierarchical group-sparsity over GeZIGs, which is further utilized to construct a compact sub-network as presented in Figure 3c. As shown in Table 1, compared to the super-network, the sub-network utilizes 54% of parameters and 51% of FLOPs to achieve a Top-1 validation accuracy 84.7% which is negligibly lower than the super-network by 0.2%.

**StackedUnets on SVNH.** We then consider a StackedUnets over SVNH. The StackedUnets is constructed by stacking two standard Unets (Ronneberger et al., 2015) with different down-samplers together, as depicted in Figure 5a in Appendix C. We employ OTOv3 to automatically build the dependency graph, establish the search space, and train by H2SPG. H2SPG identifies and projects the redundant structures onto zero and optimize the remaining important ones to attain excellent performance. As displayed in Figure 5c, the right-hand-side Unet is disabled due to `node-72-node-73-node-74-node-75` being zero.[2] The path regarding the deepest depth for the left-hand-side Unet, *i.e.*, `node-13-node-14-node-15-node-19`, is marked as redundant as well. The results by OTOv3 indicate that the performance gain brought by either composing multiple Unets in parallel or encompassing deeper scaling paths is not significant. OTOv3 also validates the human design since a single Unet with properly selected depths have achieved remarkable success in numerous applications (Ding et al., 2022; Weng et al., 2019). Furthermore, as presented in Table 1, the sub-network built by OTOv3 uses 0.37M parameters and 115M FLOPs which is noticeably lighter than the full StackedUnets meanwhile significantly outperforms it by 0.8% in validation accuracy.

**DARTS (8-Cells) on STL-10.** We next employ OTOv3 on DARTS over STL-10. DARTS is a complicated super-network consisting of iteratively stacking multiple cells (Liu et al., 2018). Each cell is constructed by spanning a graph wherein every two nodes are connected via multiple operation candidates. STL-10 is an image dataset for the semi-supervising learning, where we conduct the experiments by using its labeled samples. DARTS has been well explored in the recent years. However, the existing NAS methods studied it based on a *handcrafted* search space beforehand to *locally* pick up one or two important operations to connect every two nodes. We now employ OTOv3 on an eight-cells DARTS to *automatically* establish its search space, then utilize H2SPG to one shot train it and search important structures *globally* as depicted in Figure 6c of Appendix C. Afterwards, a sub-network is automatically constructed as drawn in Figure 6d of Appendix C. Quantitatively, the sub-network outperforms the full DARTS in terms of validation accuracy by 0.5% by using only about 15%-20% of the parameters and the FLOPs of the original super-network (see Table 1).

---

[2]Recall the definition of GeZIG, if one GeZIG equals to zero, its output would be always zero given whatever inputs. Therefore, `node-72-node-73-node-74-node-75` only produces zero output even if its ancestor vertices may have non-zero parameters. As a result, the right-hand-side Unet is completely disabled.

**SuperResNet on CIFAR10.** Later on, we switch to a ResNet search space as Zen-NAS (Lin et al., 2021), referred to as SuperResNet. SuperResNet is constructed by stacking several super-residual blocks with varying depths. Each super-residual blocks contain multiple `Conv` candidates with kernel sizes as `3x3`, `5x5` and `7x7` separately in parallel (see Figure 7a). We then employ OTOv3 to one-shot automatically produce two sub-networks with 1M and 2M parameters. As displayed in Table 2, the 1M sub-network by OTOv3 outperforms the counterparts reported in (Lin et al., 2021) in terms of search cost (on an NVIDIA A100 GPU) due to the efficient single-level optimization. The 2M sub-network could reach the benchmark over 97% validation accuracy. Remark here that OTOv3 and ZenNAS use networks of fewer parameters to achieve competitive performance to the DARTS benchmarks. This is because of the extra data-augmentations such as MixUp (Zhang et al., 2017) on this experiment by ZenNAS, so as OTOv3 to follow the same training settings.

Table 2: OTOv3 over SuperResNet on CIFAR10.

| Architecture | Top-1 Acc (%) | # of Params (M) | Search Cost (GPU days) |
|---|---|---|---|
| Zen-Score-1M(Lin et al., 2021) | 96.2 | 1.0 | 0.4 |
| Synflow† (Tanaka et al., 2020) | 95.1 | 1.0 | 0.4 |
| NASWOT† (Mellor et al., 2021) | 96.0 | 1.0 | 0.5 |
| Zen-Score-2M(Lin et al., 2021) | 97.5 | 2.0 | 0.5 |
| SANAS-DARTS (Hosseini & Xie, 2022) | 97.5 | 3.2 | 1.2* |
| ISTA-NAS(He et al., 2020) | 97.5 | 3.3 | 0.1 |
| CDEP (Rieger et al., 2020) | 97.2 | 3.2 | 1.3* |
| DARTS (2nd order) (Liu et al., 2018) | 97.2 | 3.1 | 1.0 |
| PrDARTS (Zhou et al., 2020) | 97.6 | 3.4 | 0.2 |
| P-DARTS (Chen et al., 2019) | 97.5 | 3.6 | 0.3 |
| PC-DARTS (Xu et al., 2019) | 97.4 | 3.9 | 0.1 |
| **OTOv3**-SuperResNet-1M | 96.3 | 1.0 | 0.1 |
| **OTOv3**-SuperResNet-2M | 97.5 | 2.0 | 0.1 |

† Reported in (Lin et al., 2021).
* Numbers are approximately scaled based on (Hosseini & Xie, 2022).

Table 3: OTOv3 over DARTS on ImageNet and comparison with state-of-the-art methods.

| Architecture | Test Acc. (%) | | # of Params (M) | FLOPs (M) | Search Method |
|---|---|---|---|---|---|
| | Top-1 | Top-5 | | | |
| Inception-v1 (Szegedy et al., 2015) | 69.8 | 89.9 | 6.6 | 1448 | Manual |
| ShuffleNet 2× (v2) (Ma et al., 2018) | 74.9 | – | 5.0 | 591 | Manual |
| NASNet-A (Zoph et al., 2018) | 74.0 | 91.6 | 5.3 | 564 | RL |
| MnasNet-92 (Tan et al., 2019) | 74.8 | 92.0 | 4.4 | 388 | RL |
| AmoebaNet-C (Real et al., 2019) | 75.7 | 92.4 | 6.4 | 570 | Evolution |
| DARTS (2nd order) (CIFAR10) (Liu et al., 2018) | 73.3 | 91.3 | 4.7 | 574 | Gradient |
| P-DARTS (CIFAR10) (Chen et al., 2019) | 75.6 | 92.6 | 4.9 | 557 | Gradient |
| PC-DARTS (CIFAR10) (Xu et al., 2019) | 74.9 | 92.2 | 5.3 | 586 | Gradient |
| SANAS (CIFAR10) (Hosseini & Xie, 2022) | 75.2 | 91.7 | – | – | Gradient |
| ProxylessNAS (ImageNet) (Cai et al., 2018) | 75.1 | 92.5 | 7.1 | 465 | Gradient |
| PC-DARTs (ImageNet) (Xu et al., 2019) | 75.8 | 92.7 | 5.3 | 597 | Gradient |
| ISTA-NAS (ImageNet) (Yang et al., 2020) | 76.0 | 92.9 | 5.7 | 638 | Gradient |
| **OTOv3** on DARTS (ImageNet) | 75.3 | 92.5 | 4.8 | 547 | Gradient |

(CIFAR10) / (ImageNet) refer to using either CIFAR10 or ImageNet for searching architecture.

**DARTS (14-Cells) on ImageNet.** We finally present the benchmark DARTS super-network stacked by 14 cells on ImageNet. We employ OTOv3 over it to automatically figure out the search space which the code base required specified handcraftness in the past, train by H2SPG to figure out redundant structures, and construct a sub-network as depicted in Figure 8d. Quantitatively, we observe that the sub-network produced by OTOv3 achieves competitive top-1/5 accuracy compared to other state-of-the-arts as presented in Table 3. Remark here that it is *engineeringly* difficult yet to inject architecture variables and build a multi-level optimization upon a search space being automatically constructed and globally searched. The single-level H2SPG does not leverage a validation set as others to favor the architecture search and search over the operations without trainable variables, *e.g.*, skip connection, consequently the achieved accuracy does not outperform PC-DARTS and ISTA-NAS. We leave further accuracy improvement based on the *automatic* search space as future work.

## 5   Conclusion

We propose the third generation of Only-Train-Once framework (OTOv3). To the best of knowledge, OTOv3 is the first automated system that automatically establishes the search spaces for general super-networks, then trains the super-networks via a novel H2SPG optimizer in the one-shot manner, finally automatically produces compact sub-networks of high-performance. Meanwhile, H2SPG is also perhaps the first stochastic optimizer that effectively solve a hierarchical structured sparsity problem for deep learning tasks. OTOv3 further significantly reduces the human efforts upon the existing NAS works, opens a new direction and establishes benchmarks regarding the automated NAS for the general super-networks which currently require numerous handcraftness beforehand.

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
