# A  Implementation Details

We provide more implementation details of OTOv3 library and experiments. For review purpose, we provide a library snapshot at `https://tinyurl.com/otov3-review`. The official version along with documentations and tutorials will be released to the public after the review process .

## A.1  Library Implementations

**Overview.**  OTOv3 is built upon the OTOv2 library[3] that enables automatic general sup-network training and searching sub-networks in the one-shot manner. Up to the present, the implementation of OTOv3 depends on PyTorch and ONNX (`https://onnx.ai`). ONNX is used to obtain the trace graph and the sub-network by modifying the super-network in ONNX format. H2SPG is developed as an instance of the PyTorch optimizer class. As a fundamental AI infrastructure, OTOv3 makes a significant breakthrough in AutoML to first enable the search of sub-networks from training general super-networks. Further progress and contributions from both our team and the wider open-source community are necessary to sustain its continued success.

**Limitations.**  The current version of the library relies on ONNX, which means that the super-networks need to be convertible into the ONNX format. Meanwhile, if the super-network contains unsupported operators, the library may not function normally. To address this, we are committed to maintaining and adding new operators to the library, and leverage contributions from the open-source community in this regard. Additionally, we are actively working on reducing the dependency on ONNX to broaden the library's coverage and compatibility.

Furthermore, to prioritize generality, we avoid requiring users to manually introduce auxiliary architecture variables, as seen in the existing gradient-based NAS methods. To search without architecture variables, the current OTOv3 library formulates a hierarchical structured sparsity optimization to identify redundant minimal removal structures based on sparse optimization. We currently require the minimal removal structures to have trainable variables. Consequently, the operations without trainable variables such as `skip connection` are not removal for the current version of OTOv3 yet. Identifying and removing operations without trainable variables is an aspect that we consider as future work and plan to address in subsequent updates.

## A.2  Experiment Implementations

All experiments were conducted on an NVIDIA A100 GPU. The search cost of OTOv3 was calculated as the runtime of the warm-up phase in Algorithm 3, since it is during this phase that the redundant group candidates are constructed. In our experiments, H2SPG follows the existing NAS works (Liu et al., 2018) by performing 50 epochs for architecture search during the warm-up phase and evolving the learning rate using a cosine annealing scheduler.

For the SuperResNet experiments, we adopt the data augmentation technique of MixUp, following the training settings of ZenNAS (Lin et al., 2021), and employ a multiple-period cosine annealing scheduler. The maximum number of epochs for the DemoSupNet and StackedUnets is set to 300 following (Chen et al., 2023). In the case of DARTS on ImageNet, we expedite the training process by constructing a sub-network once the desired redundant group sparsity level is reached. We then train this sub-network until convergence. All other experiments are carried out in the one-shot manner.

The initial learning rate is set to 0.1 for most experiments, except for the DARTS experiments where it is set to 0.01. The lower initial learning rate in DARTS is due to the absence of auxiliary architecture variables in our super-network, which compute a weighted sum of outputs. Additionally, operations without trainable variables, such as `skip connections`, are preserved (refer to the limitations). Consequently, the cosine annealing period is repeated twice for the DARTS experiments to account for the smaller initial learning rate. The mini-batch sizes are selected as 64 for all tested datasets, except for ImageNet, where it is set to 128. The target group sparsities are estimated in order to achieve a comparable number of parameters to other benchmarks. This is accomplished by randomly selecting a subset of GeZIGs to be zero and then calculating the parameter quantities in the constructed sub-networks. The remaining hyper-parameter settings of H2SPG adhere to the default settings of DHSPG in OTOv2.

---

[3] `https://tinyurl.com/only-train-once`

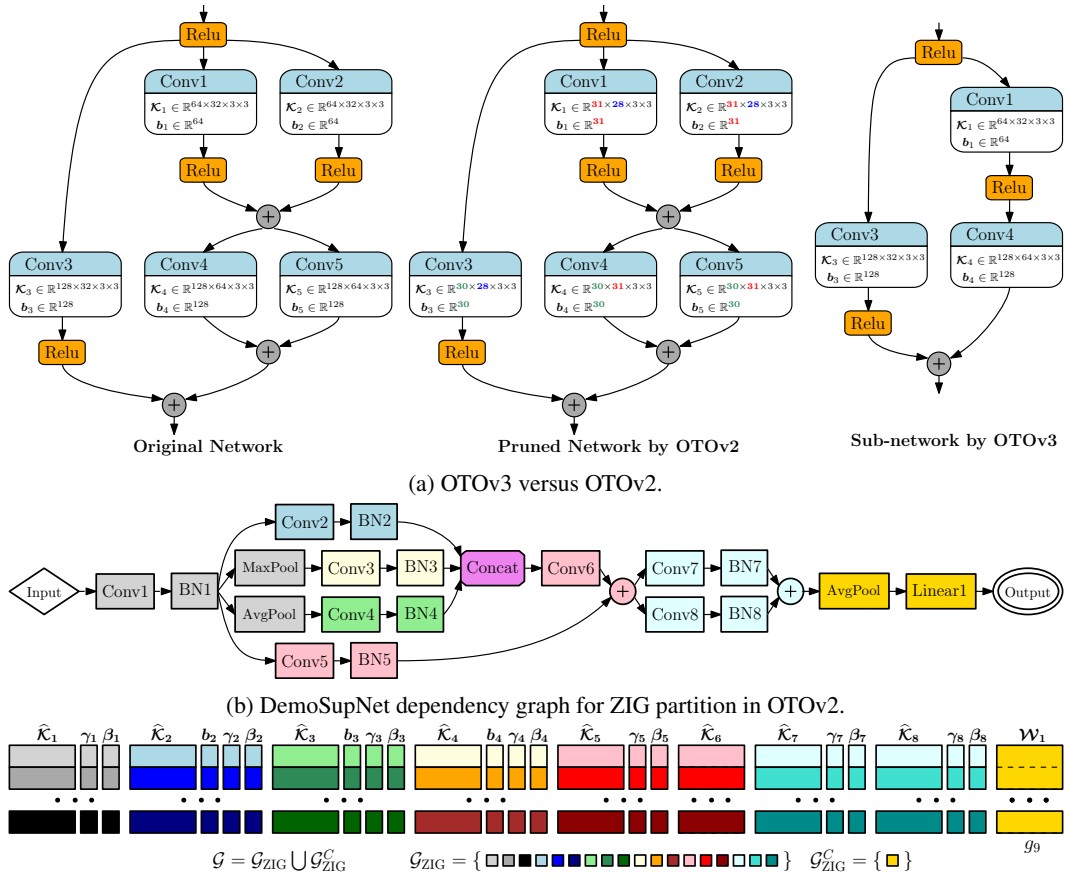

(a) OTOv3 versus OTOv2.

(b) DemoSupNet dependency graph for ZIG partition in OTOv2.

(c) DemoSupNet ZIG partition in OTOv2.

Figure 4: (a) OTOv3 versus OTOv2. (b) Dependency graph for DemoSupNet by OTOv2 for automatic structured pruning. (c) ZIG partition for DemoSupNet by OTOv2.

## B  ZIG versus GeZIG

We aim to clarify the distinction between the Zero-Invariant Group (ZIG) introduced in OTOv2 (Chen et al., 2023) and the Generalized Zero-Invariant Group (GeZIG) proposed in this work. Broadly speaking, ZIG and GeZIG refer to different categories of minimal removal structures satisfying zero-invariant property, serving distinct purposes to search optimal architecture within the super-network.

A minimal removal structure refers to a structure that satisfies two conditions: *(i)* removing it from the deep neural network (DNN) still leaves a valid network, and *(ii)* the structure cannot be further decomposed into smaller removal structures, making it minimal. The specific forms of minimal removal structures depend on whether the vertices (representing operations) and edges (representing connections) in the trace graph are preserved or not.

ZIG (Zero-Invariant Group) pertains to the minimal removal structure employed in the context of structure pruning. In structure pruning, the vertices (operations) and edges (connections) of the trace graph $(\mathcal{V}, \mathcal{E})$ are preserved, but the vertices become slimmer. In contrast, GeZIG (Generalized Zero-Invariant Group) focuses on the standard NAS[4] problem that involves removing redundant vertices and edges entirely. GeZIG considers the entire removal of vertices and edges in order to search for more compact and efficient network architectures.

Figure 4a presents the difference between OTOv3 and OTOv2, which train the super-networks and search compact sub-networks from different perspectives. Consequently, the dependency graph for the GeZIG partition differs significantly from the dependency graph for the ZIG partition.

---

[4]Structured pruning actually can be interpreted as NAS as well, *i.e.*, searching inside each operation.

As illustrated Figure 4b, which shows the example of dependency graph for ZIG over DemoSupNet, the connected components marked with the same color represent vertices that have affiliations and need to be pruned together. For instance, the output tensors of `Conv5-BN5` and `Conv6` are added together, requiring them to have the same shapes for the addition operation. Therefore, both `Conv5-BN5` and `Conv6` need to be slimmed by removing the same number of filters, bias or weight scalars. Consequently, their filters, bias scalar, BN bias, and weight bias are grouped together as a single ZIG, as depicted in Figure 4c.

In contrast, GeZIG groups the trainable variables of several entire vertices together, as shown in Figure 1c. Moreover, GeZIG needs to consider the hierarchy of the dependency graph to ensure the validity of the sub-network since entire connections and vertices are removed. This hierarchical consideration is necessary for NAS and brings significant challenges to the structured sparse optimizers.

## C  Graph Visualizations

In this appendix, we present visualizations generated by the OTOv3 library to provide more intuitive illustrations of the architectures tested in the paper. The visualizations include trace graphs, dependency graphs, identified redundant GeZIGs, and constructed sub-networks. To ensure clear visibility, we highly recommend **zooming in with an upscale ratio of at least 500%** to observe finer details and gain a better understanding of the proposed system.

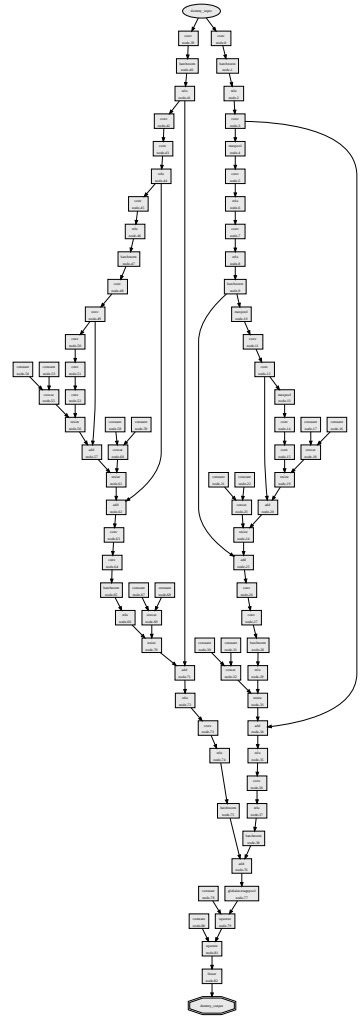

(a) StackedUnets trace graph.

Figure 5: StackedUnets illustrations drawn by OTOv3.

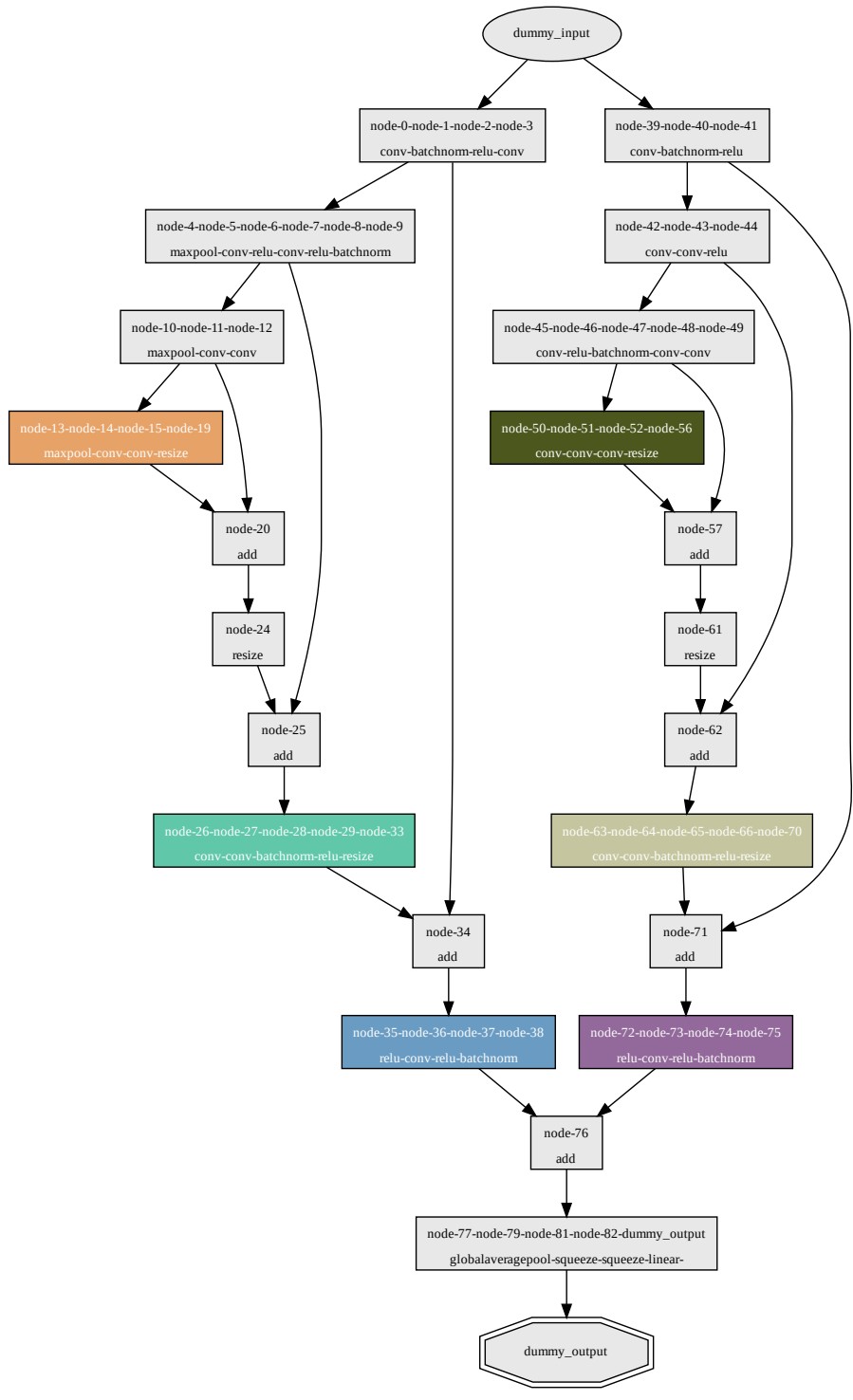

(b) StackedUnets search space.

Figure 5: StackedUnets illustrations drawn by OTOv3.

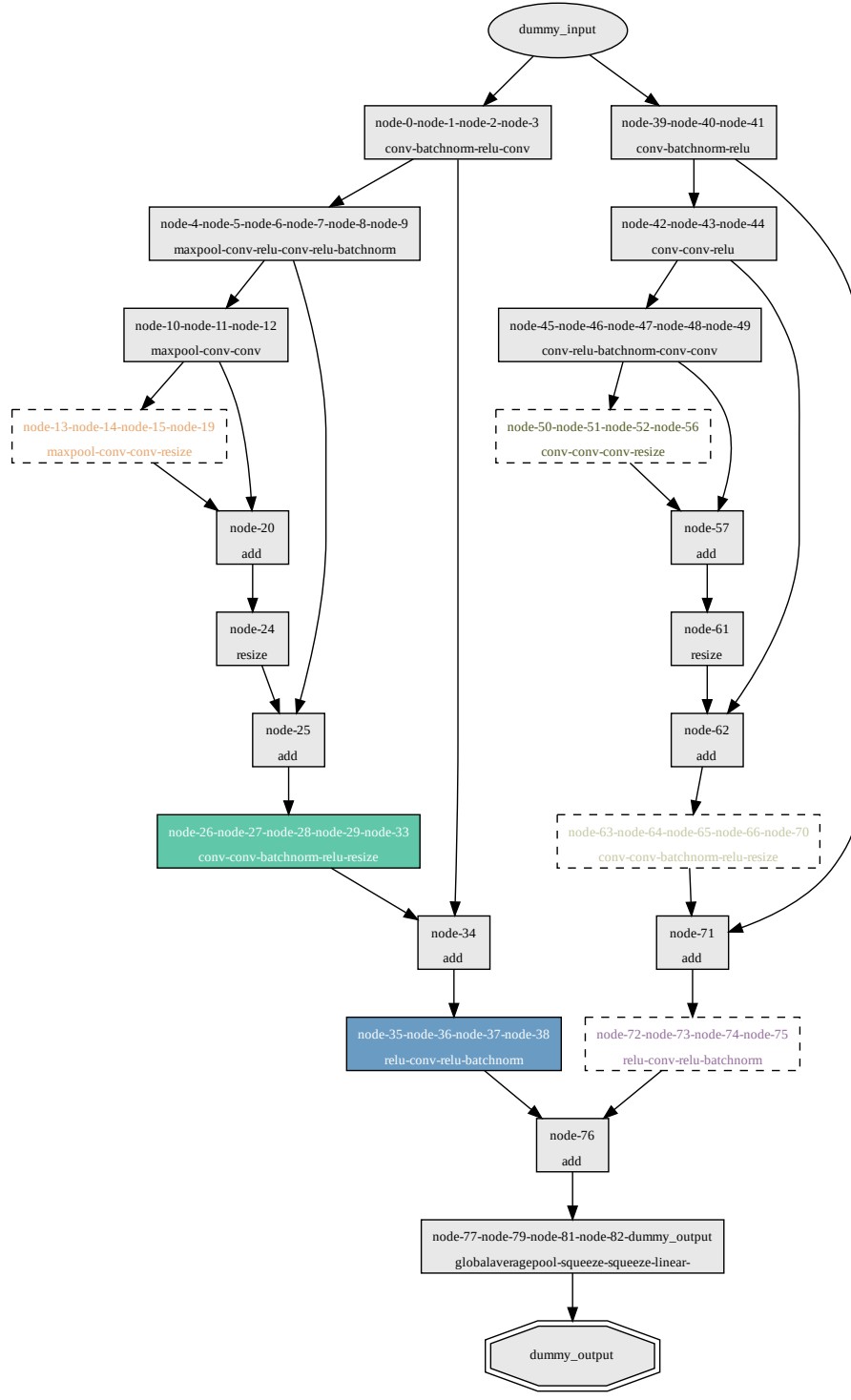

(c) StackedUnets dependency graph with identified removal vertices.

Figure 5: StackedUnets illustrations drawn by OTOv3.

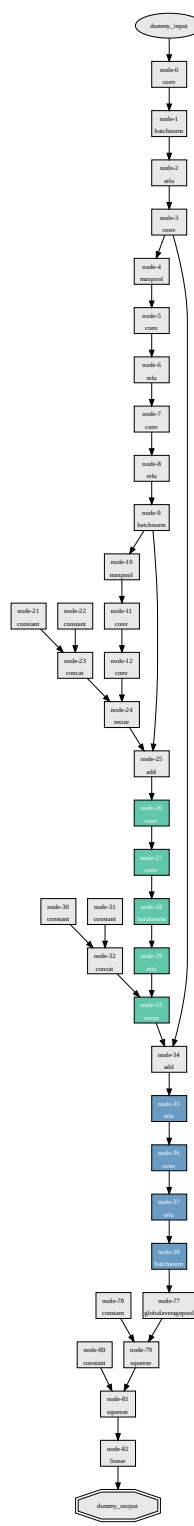

(d) Constructed sub-network upon StackedUnets.

Figure 5: StackedUnets illustrations drawn by OTOv3.

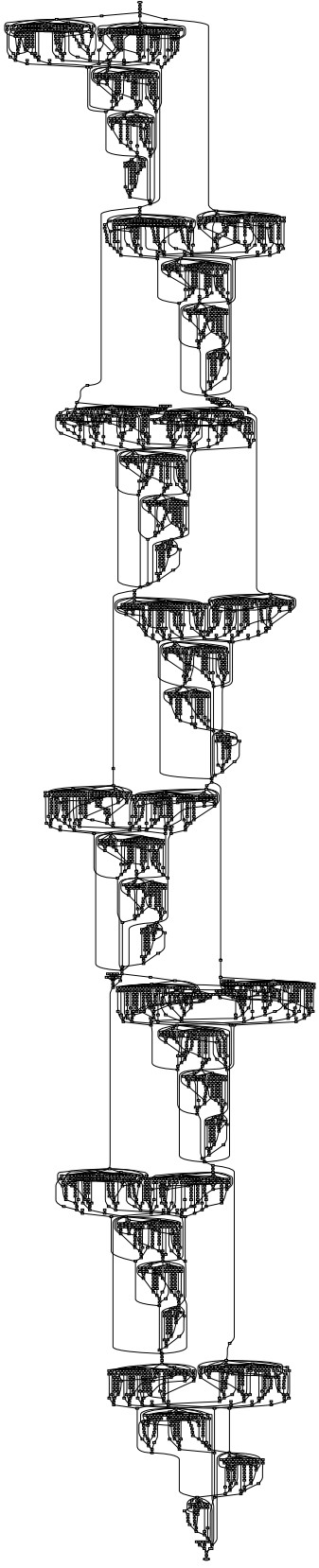

(a) DARTS (8 cells) trace graph.

Figure 6: DARTS (8 cells) illustrations drawn by OTOv3.

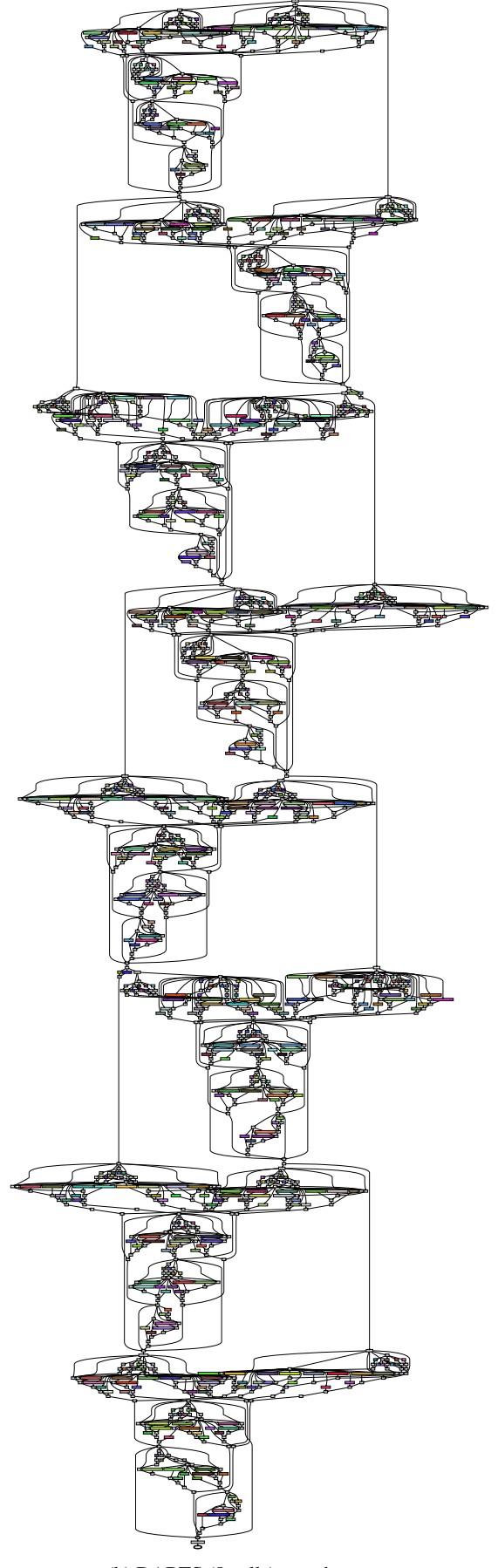

(b) DARTS (8 cells) search space.

Figure 6: DARTS (8 cells) illustrations drawn by OTOv3.

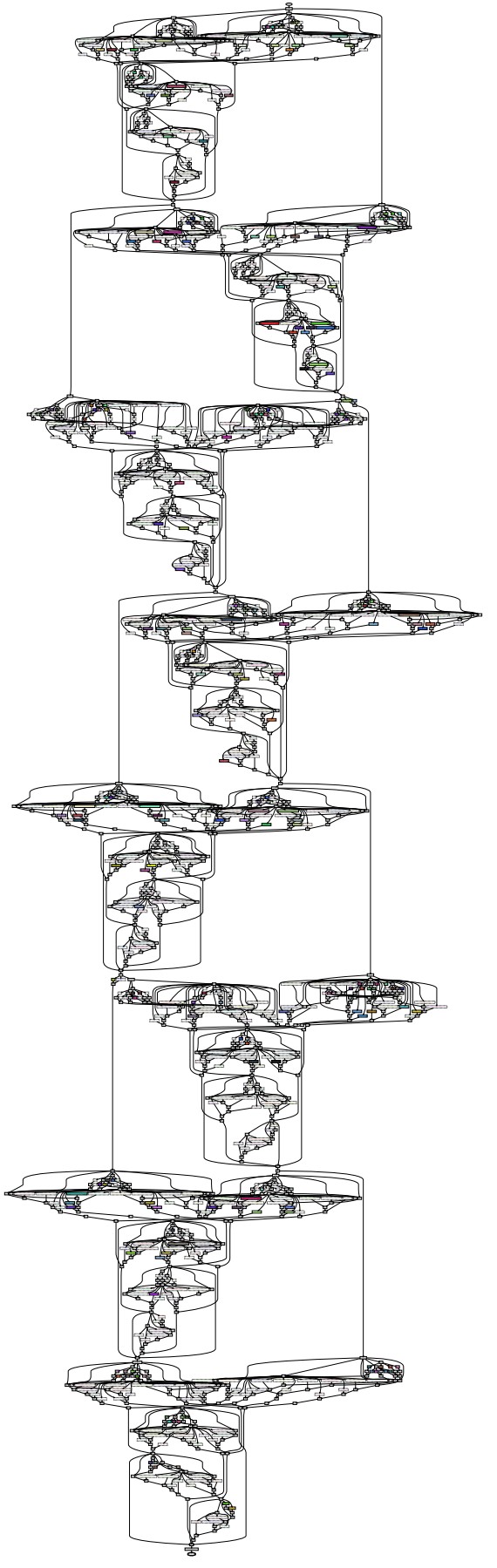

(c) DARTS (8 cells) dependency graph with identified removal vertices.

Figure 6: DARTS (8 cells) illustrations drawn by OTOv3.

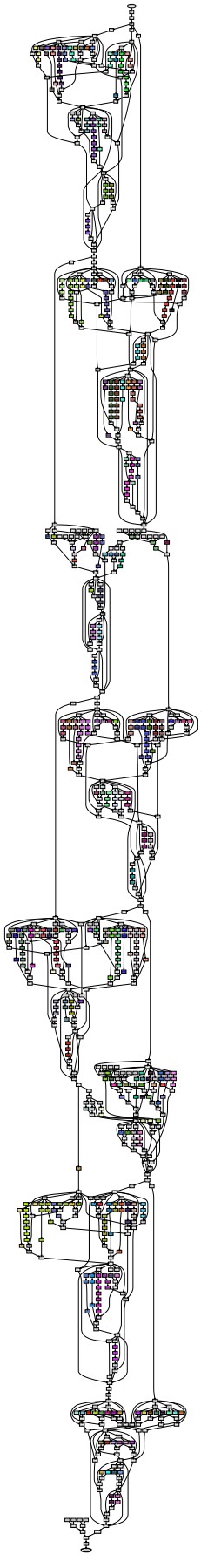

(d) Constructed sub-network upon DARTS (8 cells).

Figure 6: DARTS (8 cells) illustrations drawn by OTOv3.

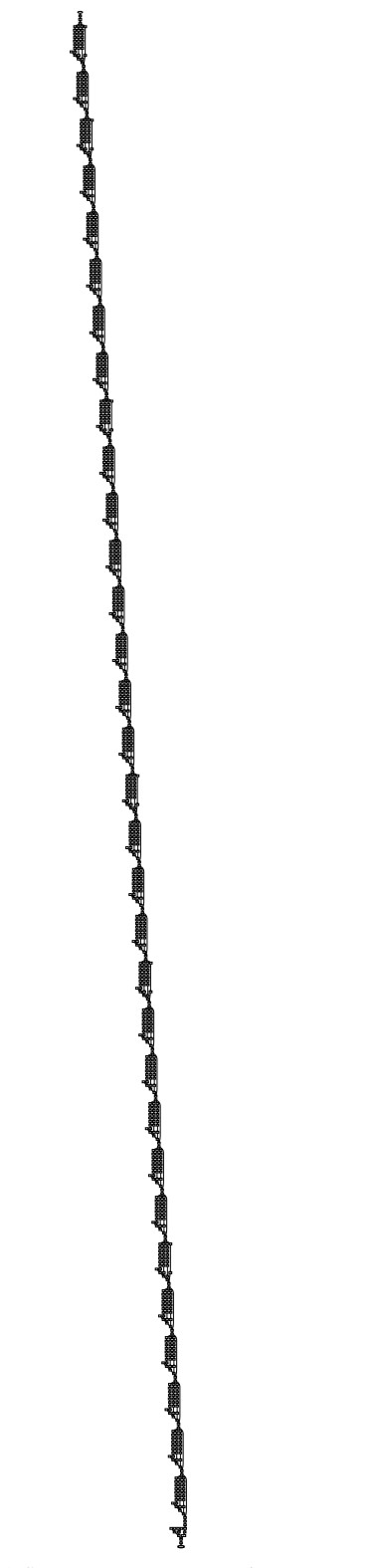

(a) SuperResNet trace graph.

Figure 7: SuperResNet illustrations drawn by OTOv3.

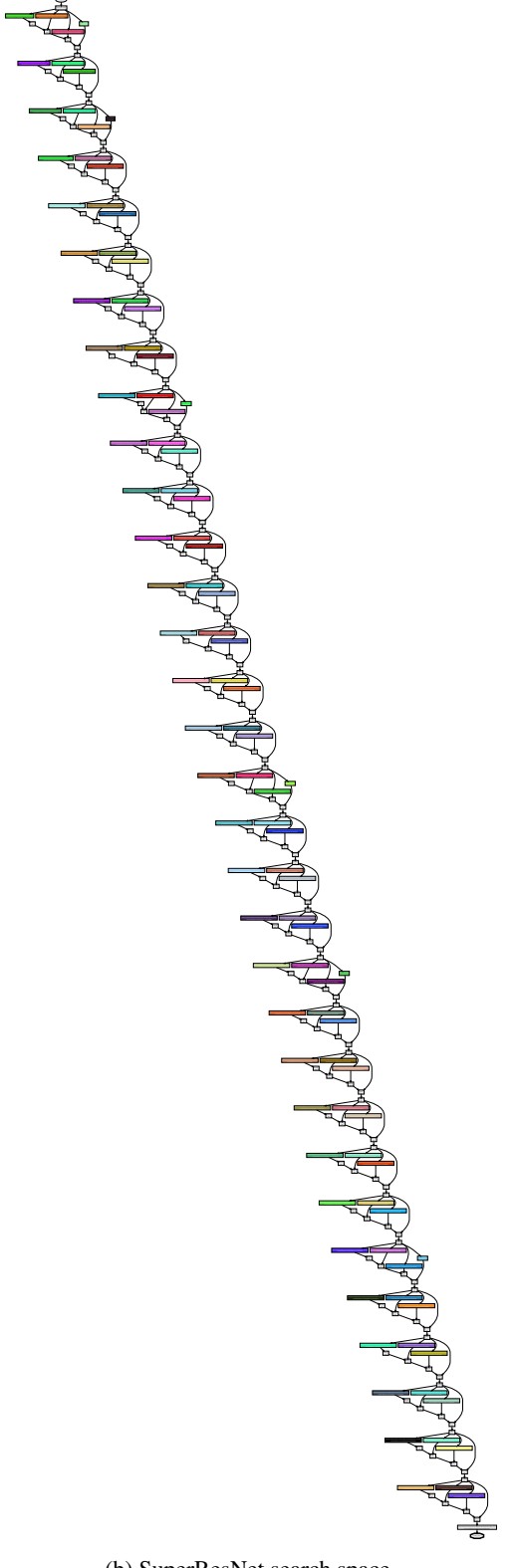

(b) SuperResNet search space.

Figure 7: SuperResNet illustrations drawn by OTOv3.

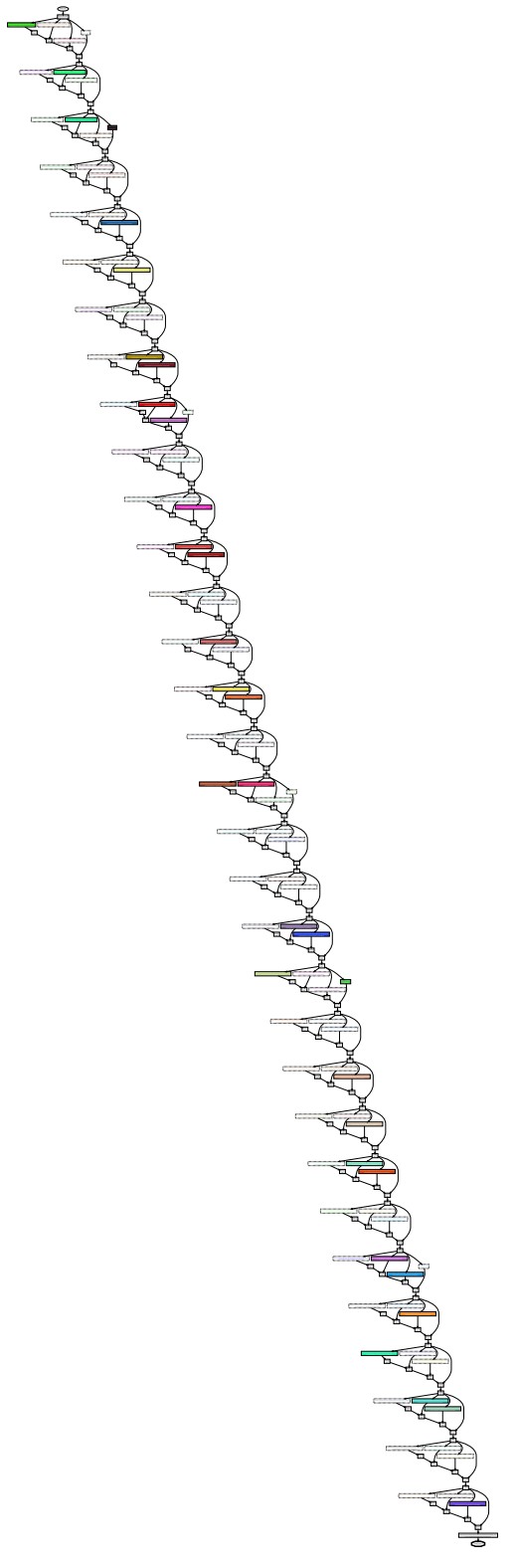

(c) SuperResNet dependency graph with identified removal vertices.

Figure 7: SuperResNet illustrations drawn by OTOv3.

(d) Constructed sub-network upon SuperResNet.

Figure 7: SuperResNet illustrations drawn by OTOv3.

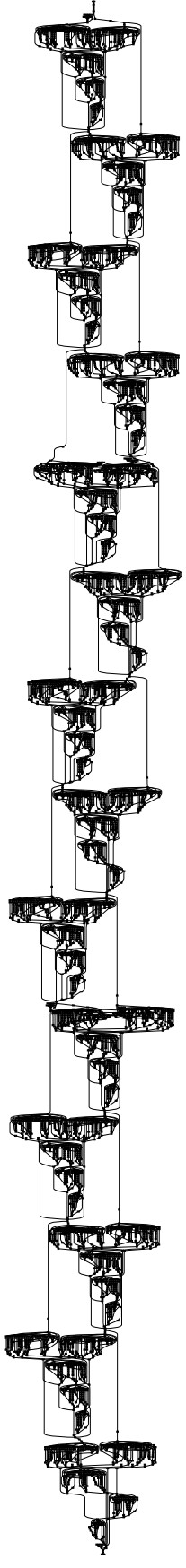

(a) DARTS (14 cells) trace graph.

Figure 8: DARTS (14 cells) illustrations drawn by OTOv3.

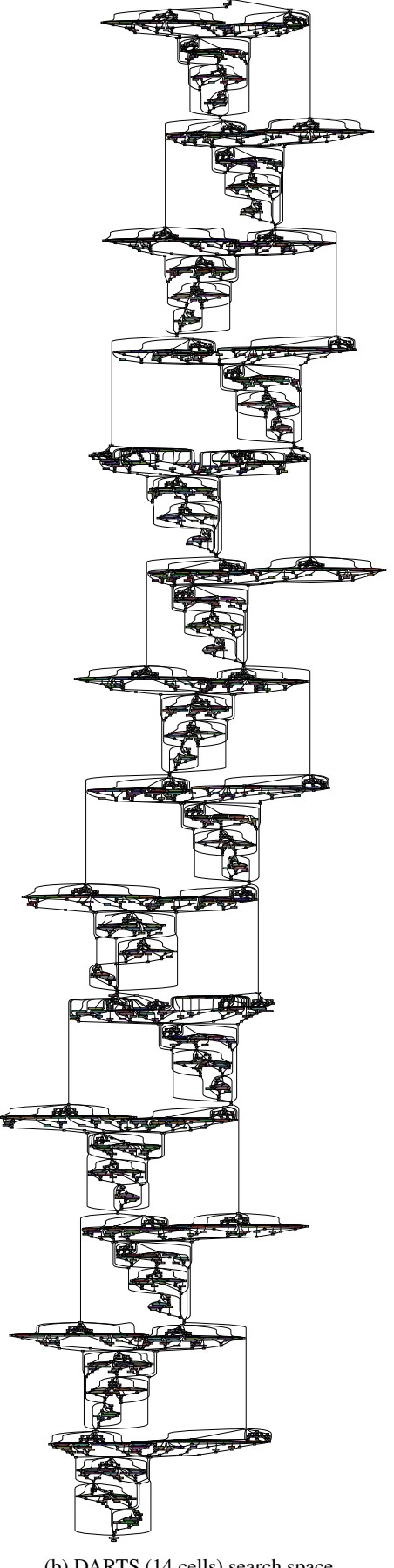

(b) DARTS (14 cells) search space.

Figure 8: DARTS (14 cells) illustrations drawn by OTOv3.

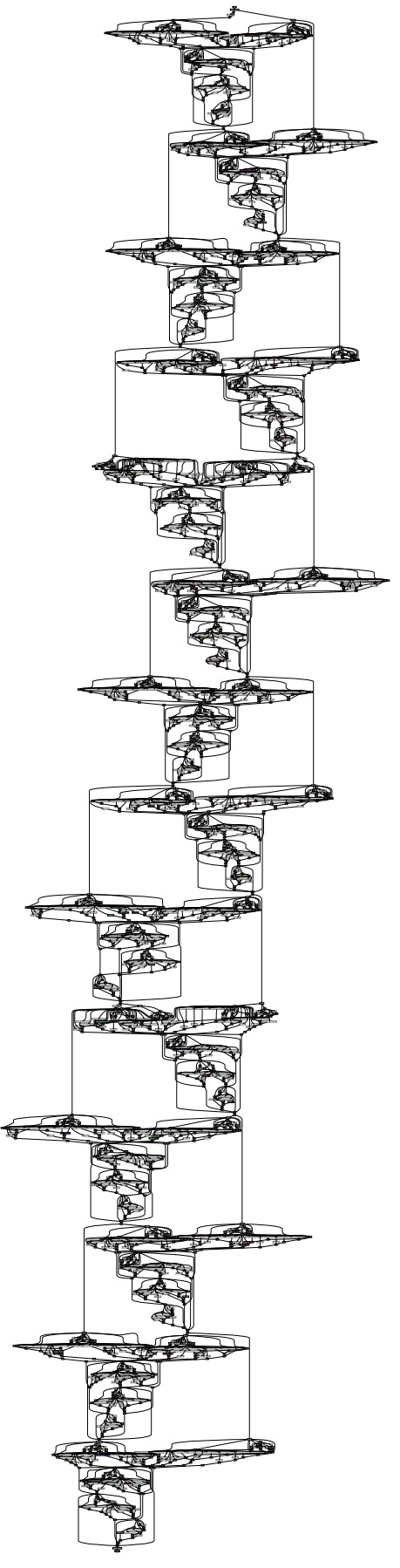

(c) DARTS (14 cells) dependency graph with identified removal vertices.

Figure 8: DARTS (14 cells) illustrations drawn by OTOv3.

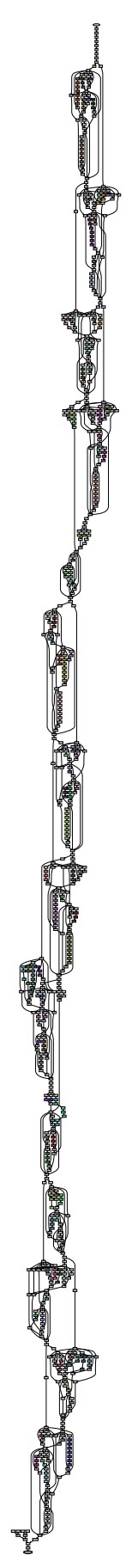

(d) Constructed sub-network upon DARTS (14 cells).

Figure 8: DARTS (14 cells) illustrations drawn by OTOv3.

## D  Ablation Studies Extensive Experiments

In this appendix, we conduct extensive experiments and ablation studies over one more benchmark architecture RegNet (Radosavovic et al., 2020) on CIFAR10. Without loss of generality, we employ OTOv3 over the RegNet-800M which has accuracy 95.01% on CIFAR10 according to `https://github.com/yhhhli/RegNet-Pytorch`. As other experiments, OTOv3 automatically constructs its search space, trains via H2SPG, and establishes the sub-networks without fine-tuning.

We conduct ablations by employing OTOv3 with two sparse optimizers: H2SPG and DHSPG. We separately evaluate them with varying target hierarchical group sparsity levels in problem (1) across a range of $\{0.1, 0.3, 0.5, 0.7, 0.9\}$. The obtained results are from three independent tests initialized with different random seeds, and reported in Table 4.

Table 4: OTOv3 on RegNet on CIFAR10.

| Backend | Method | Optimizer | Target Group Sparsity | # of Params (M) | Top-1 Acc. (%) |
|---------|--------|-----------|-----------------------|-----------------|----------------|
| RegNet-200M | Baseline | SGD | – | 2.31 | 93.58 |
| RegNet-400M | Baseline | SGD | – | 4.77 | 94.15 |
| RegNet-600M | Baseline | SGD | – | 5.67 | 94.73 |
| RegNet-800M | Baseline | SGD | – | 6.60 | 95.01 |
| RegNet-800M | **OTOv3** | DHSPG | 0.1 | $5.56 \pm 0.02$ | $95.26 \pm 0.13$ |
| RegNet-800M | **OTOv3** | DHSPG | 0.3 | $(3.40, ✗, ✗)$ | $(95.01, ✗, ✗)$ |
| RegNet-800M | **OTOv3** | DHSPG | 0.5 | $(✗, ✗, ✗)$ | $(✗, ✗, ✗)$ |
| RegNet-800M | **OTOv3** | DHSPG | 0.7 | $(✗, ✗, ✗)$ | $(✗, ✗, ✗)$ |
| RegNet-800M | **OTOv3** | DHSPG | 0.9 | $(✗, ✗, ✗)$ | $(✗, ✗, ✗)$ |
| RegNet-800M | **OTOv3** | **H2SPG** | 0.1 | $5.58 \pm 0.01$ | $95.30 \pm 0.10$ |
| RegNet-800M | **OTOv3** | **H2SPG** | 0.3 | $3.54 \pm 0.15$ | $95.08 \pm 0.14$ |
| RegNet-800M | **OTOv3** | **H2SPG** | 0.5 | $1.83 \pm 0.09$ | $94.61 \pm 0.19$ |
| RegNet-800M | **OTOv3** | **H2SPG** | 0.7 | $1.16 \pm 0.12$ | $91.92 \pm 0.24$ |
| RegNet-800M | **OTOv3** | **H2SPG** | 0.9 | $0.82 \pm 0.17$ | $87.91 \pm 0.32$ |

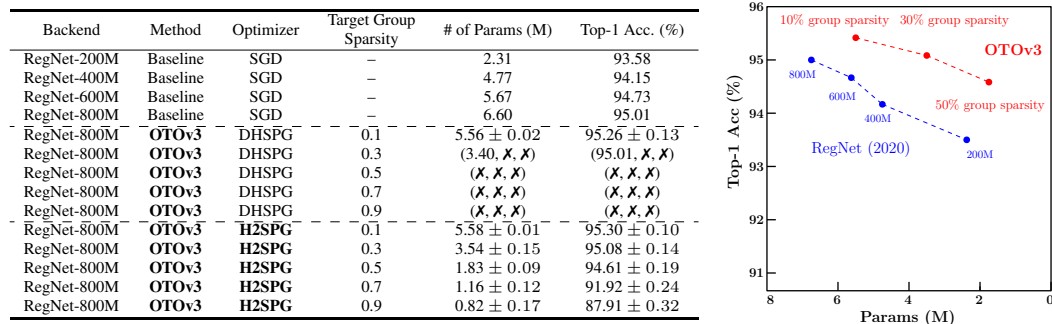

**Sub-networks by OTOv3 versus Super-Networks.**  As presented in Table 4, the sub-networks under varying hierarchical group sparsity levels computed by OTOv3 with H2SPG exhibits the Pareto frontier comparing with the benchmark RegNet. Notably, the sub-networks under target group sparsity levels of 0.1 and 0.3 outperform the full RegNet-800M. Furthermore, the sub-network produced with a group sparsity level of 0.5 outperforms the RegNet200M, RegNet400M, and RegNet600M, despite utilizing significantly fewer parameters while achieving higher accuracy.

**H2SPG versus DHSPG.**  In Table 4, a comparison between H2SPG and DHSPG reveals that DHSPG often fails when confronts with reasonably large target sparsity levels. The failure tests are denoted by the symbol ✗. The underlying reason lies in its design, which solely treats problem (1) as an independent and disjoint structured sparsity problem. By disregarding the hierarchy within the network, DHSPG easily generates sub-networks that lack validity. Conversely, H2SPG takes into account the network hierarchy and successfully addresses the target problem (1). This stark contrast highlights the superior performance of H2SPG in producing valid sub-networks via identifying and removing entire redundant operations and connections.

## E  Complexity Analysis

We end the appendix via analyzing the time and space complexity in OTOv3 to construct the search space and the hierarchy consideration during H2SPG optimization.

**Search Space Construction.**  The automatic search space construction Algorithm 2 primarily a customized graph algorithm designed to identify minimal removal structures and partition trainable variables into GeZIGs. It contains two main stages: *(i)* establishing the dependency graph, and *(ii)* constructing the GeZIG partition.

During the first stage, the algorithm traverses the trace graph using a combination of depth-first and breadth-first approaches with specific operations. Consequently, the worst-case time complexity is $\mathcal{O}(|\mathcal{V}| + |\mathcal{E}|)$ to visit every vertex and edge in the trace graph. The worst-case space complexity equals to $\mathcal{O}(|\mathcal{V}|)$ due to the queue container used in Algorithm 2 and the cache employed during the recursive depth-first search. In the second stage, the constructed dependency graph $(\mathcal{V}_d, \mathcal{E}_d)$ is traversed in a depth-first manner to perform the GeZIG partition. In the worst case scenario, where

$(\mathcal{V}_d, \mathcal{E}_d)$ equals $(\mathcal{V}, \mathcal{E})$, the time and space complexities remain the same as $\mathcal{O}(|\mathcal{V}| + |\mathcal{E}|)$ and $\mathcal{O}(|\mathcal{V}|)$ respectively. In summary, the worst-case time complexity for both stages combined is $\mathcal{O}(|\mathcal{V}| + |\mathcal{E}|)$, and the worst-case space complexity is $\mathcal{O}(|\mathcal{V}|)$. Therefore, the search space construction can be typically efficiently finished in practice.

**Hierarchy Structured Sparsity Optimization.** Compared to standard structured sparsity optimizers, H2SPG takes into account of the hierarchy of the network during optimization to ensure the validity of the generated sub-networks. This is achieved through a hierarchy check, which involves removing one vertex from the dependency graph $(\mathcal{V}_d, \mathcal{E}_d)$ and determining if the remaining DNN remains connected from the input to the output. A depth-first search is performed for this purpose, with a worst-case time complexity of $\mathcal{O}(|\mathcal{V}_d| + |\mathcal{E}_d|)$ and a worst-case space complexity of $\mathcal{O}(|\mathcal{V}_d|)$.

Throughout the optimization process, the hierarchy check is only triggered once iteratively over a subset of minimal removal structures (proportional to the target group sparsity level). This check occurs immediately after the warm-up phase in Algorithm 3. Consequently, the worst-case overall time complexity for the hierarchy check is $\mathcal{O}(|\mathcal{V}_d|^2 + |\mathcal{E}_d| \cdot |\mathcal{V}_d|)$. The worst-case overall space complexity remains $\mathcal{O}(|\mathcal{V}_d|)$, since the cache used for the hierarchy check is cleaned up after each vertex completes its own check.

It is important to note that although the worst-case time complexity is quadratic in the number of vertices of the constructed dependency graph, the hierarchy check can be *efficiently* executed in practice because the number of vertices in the dependency graph is typically reasonably limited. Additionally, the hierarchy check only occurs *once* during the entire optimization process, consequently does not bring significant computational overhead to the whole process.