# OpenReview forum: "OTOv3: Towards Automatic Sub-Network Search Within General Super Deep Neural Networks"
_NeurIPS.cc/2023/Conference — Submitted to NeurIPS 2023_

### Official Review · Reviewer_bA4X · 2023-07-01

**Soundness:** 2 fair
**Presentation:** 2 fair
**Contribution:** 2 fair
**Rating:** 4
**Confidence:** 5

**Summary:**

OTOv3 is an automated system that trains general super-networks without pretraining or fine-tuning, constructs search spaces automatically, and produces high-performing sub-networks. Experimental results show competitive or superior performance compared to state-of-the-art methods across different benchmark datasets and architectures.

**Strengths:**

Strengths are shown below:

1. OTOv3 with codebase sounds like a systematic work, which may have some actual impacts.

2. Exquisite drawing is appealing.

**Weaknesses:**

However, some weaknesses make me think that I am not fully agreeing with this article now.

1. Although I have read the comparison in the appendix, I find that the work lacks novelty when compared to OTOv2. It appears that some key modules in OTOv3 are simply improved versions of those in OTOv2. For example, the comparison between GeZIGs and ZIG, Dependency Graph Construction in OTOv2 and H2SPG versus DHSPG. As a result, OTOv3 does not seem to introduce any essential innovations but rather represents a slight technological improvement over OTOv2.

2. Some of the comparisons made in the experiments are unfair. In Table 2, the search cost is not fairly assessed since the author's value was obtained using a new GPU (A100), while others were conducted using an older version GPU. Additionally, the authors fail to consider the training time of the super-Net when comparing it with Training-free methods. In Table 3, some of the baselines used are outdated, not the most recent ones. The authors mainly compare papers published before 2020, with only one paper from 2022. A more appropriate reference would be OTOv2.

3. The methods sections are poorly organized. Even after reading the Method section multiple times, I am still confused about how this method actually works. The authors often introduce concepts without providing a sufficient explanation in the appropriate positions, resulting in explanations being scattered throughout the text, which is highly confusing. It seems as though the authors assume the readers are already familiar with these concepts. For instance, "$x^*_{H2SPG}$" is first mentioned in the introduction but not adequately explained.

4. The writing itself needs further improvement. For example, the sentence in line 39 ends abruptly, and it is unclear whether the subsequent sentences in line 40 should be read as a continuation. In the introduction, the authors use the phrase "perhaps the first" to describe their method. If the authors are confident that it is indeed the "first," they should remove the word "perhaps" or refrain from using "the first" altogether.

**Questions:**

The authors could refer to the Weaknesses. Additionally, I also have some open questions to discuss.

1. It seems that the structures searched by the OTOv3 are complicated. So could the authors provide a comparison of inference time in Table 3?

2. Could the authors reveal the insights behind the methods with few words?

---

> ### Author Rebuttal · Authors · 2023-08-04
>
> Dear Reviewer bA4X,
>
> We appreciate your constructive comments and valued suggestions. Please see our responses as follows. Look forward to further discussion.
>
> - **Lacks novelty compared with OTOv2**.
>
> > Thank you for the question. Please refer to our general response above along with a PDF to illustrate the difference between OTOv3 and OTOv2. We hope that could adequately address your concern.
>
> - **Search cost comparison with zero-shot methods are unfair.**
>
> > Thank you for the insightful question. We provide one general response above where we hope could adequately tackle your concern.
>
> - **Compare with more recent works. And why not compare with OTOv2?**
>
> > Thanks for the great suggestion. We have included more recent works into the preliminary comparison.  Considering that OTOv3 and OTOv2 focus on divergent target problems, we have elected not to include OTOv2 in our comparative experiments.
>
> >| Method | Search Space | Acc1 (%) | Params | GPU | Search Cost (GPU days) |
> >|--|--|--|--|--|--|
> >| Zen-Score-2M (2021) | ResNet Pool | 97.5 |2M | A100 | 0.5 |
> >| SANAS-DARTS (2022) | DARTS | 97.5 | 3.2M | 1080Ti | 4.6 |
> >| **ZARTS (2022)** | DARTS | 97.5 | 3.5M | 2080Ti | 1.0 |
> >| **ADARTS (2022)** |  DARTS | 97.5 | 2.9M | 3090Ti | 0.2|
> >|**EPC-DARTS (2023)**| DARTS | 97.6 | 3.2M | 3090Ti | 0.2 |
> >| OTOv3 | SuperResNet | 97.5 | 2M | A100 | 0.1 |
>
>
> > [ZARTs] ZARTS: On Zero-order Optimization for Neural Architecture Search, NeurIPS, 2022.
> >
> > [ADARTS] Partial Connection Based on Channel Attention for Differentiable Neural Architecture Search, IEEE TII, 2022.
> >
> > [EPC-DARTS] EPC-DARTS: Efficient partial channel connection for differentiable architecture search, NN, 2023.
>
> - **The DARTs sub-network searched by OTOv3 seems more complicated than others. Could the authors provide a comparison of inference time?**
>
> > This is a great observation. Please see our below responses.
>
> > **Why OTOv3's searched sub-DARTS looked more complicated than others?**
> > The current OTOv3 could automatically construct search space given general DNNs, while requires searchable structures containing  trainable variables. Consequently, operators without trainable variables such as skip connections are preserved. In contrast, other methods could utilize pre-specified architecture variables to search over those operators. Meanwhile, the automated introduction of such auxiliary architecture variables over automated constructed search space is an open problem and not achieved yet by OTOv3. Therefore, the resulted sub-network by OTOv3 on DARTS typically looked more complicated than other methods.
>
> > **Inference time comparison.** We provide the inference time comparison on an Intel(R) Core(TM) i5 CPU @ 2.60GHz. We agree that the DARTS sub-network searched by OTOv3, though has competitive FLOPs, proceed slower than others due to preservation of untrainable structures.
> > For fairness, we manipulate the DARTS super-network by adding extra trainable coefficients onto the untrainable structures to enable searching over them. The preliminary result showed that in this setting, sub-network constructed by OTOv3 could reach competitive inference time to other methods.
>
> >| Method | Search over untrainable structures | Acc1 (%) | Params (M) | FLOPs (M) | Inference Time (CPU) |
> >|--|--|--|--|--|--|
> >| DARTS | Yes | 73.3 | 4.7 | 574 | 8.6ms |
> >| ISTA-NAS | Yes |76.0 | 5.7 | 638 | 9.1ms |
> > | PC-DARTs| Yes | 75.8 | 5.3 | 597 | 9.8ms |
> > | P-DARTS | Yes | 75.6 | 4.9 | 557 | 8.0ms |
> > | ProxylessNAS | Yes | 75.1 | 7.1 | 465 | 6.2ms |
> >| **OTOv3** | **No** | 75.3 | 4.8 | 547 |  **24.2ms**  |
> >| **OTOv3** | **Yes** | 75.5 |  5.2 | 563 | **8.4ms** |
> >
> > *(Tested in ONNX format to optimize the inference efficiency on CPU with an input tensor as (1, 3, 224, 224).)*
>
> > Finally, besides the above, given an automatically constructed search space by OTOv3, automatically introducing auxiliary variables or defining some training-free mechanisms would be the most appropriate solutions in our mind to enable automatic search over the structures without trainable variables. We will leave them as future work for further study.
>
> - **Improve the writing. Reveal the insights behind the methods with few words.**
>
> > Thanks for the great suggestion. We will add more descriptive languages and illustrations into the revision to improve the writing.
>
> > In a few words, given a general DNN, OTOv3 at first automatically analyzes the dependency and constructs its search space, then employs H2SPG to identify redundant structures and train the remaining important ones, finally automatically constructs a high-performing sub-network upon the solution of H2SPG.
>
> > The below is our explanations with greater details.
> >
> > **Automated search space construction.** Given a general DNN, OTOv3 at first analyzes the dependency across varying operators and creates one dependency graph. The dependency graph is then used to figure out which groups of operators are removal so that after removing the operators, the remaining DNN is still valid and functions normally. In particular, OTOv3 considers a class of such removal structures, i.e., the generalized zero-invariant groups (GeZIGs). Therefore, the set of GeZIGs forms the search space of the given DNN.
> >
> > **H2SPG to identify redundant structures.** The next is to find out among GeZIGs, which are redundant. The underlying optimization problem is a hierarchical structured sparsity optimization problem. We then proposed H2SPG to effectively find-out redundant GeZIGs and train the important GeZIGs to high-performance.  Remark here that one redundant GeZIG refers to one redundant removal structure in the given DNN.
> >
> > **Automated sub-network construction.** In the end, upon the solution of H2SPG, we automatically construct a high-performing sub-network by removing the structures corresponding to the redundant GeZIGs. Such created sub-network does not require further fine-tuning.
>
> Sincerely,
>
> Paper 5388 Authors

---

> > ### Comment · Reviewer_bA4X · 2023-08-17
> >
> > After carefully reading the explanation of the authors, I still hold the view that there is no essential innovation over OTOv2. Thanks for the authors' work.

---

> > > ### Author Response · Authors · 2023-08-17
> > > **Thanks for your responses. Look forward to further discussion.**
> > >
> > > Dear Reviewer bA4X,
> > >
> > > Thanks for your responses and efforts.
> > >
> > > We gently remark here that the counterparts of OTOv3 should be the existing NAS works rather than OTOv2, due to their orthogonal target problems. Comparing with the existing NAS works, OTOv3 has made significant essential innovations, especially given that the primary goal of OTOv3 has not been achieved by existing solutions to the best of our knowledge.
> > >
> > > We would like to further kindly highlight that OTOv3 is neither a simple technical improvement nor a direct application onto NAS tasks upon OTOv2, since there exist systematic differences of both frameworks spanning from target problems, algorithmic designs, to engineering developments. As outlined below, the technical contributions in OTOv3 are of significant importance to our understanding.
> > >
> > > |  |  OTOv3 | OTOv2|
> > > |--|--|--|
> > > | **Dependency Graph** |  Designed for Automated Sub-network Search | Designed for Automated Structured Pruning |
> > > | **Sparse Optimizer** | Consider DNN hierarchy | No hierarchy |
> > > | **Produced Network** | Dramatically change topology of computational graph | Preserve topology yet slim operators |
> > > | **Engineering Development** | The code base has thousands-of-lines difference. | -- |
> > >
> > > We would appreciate if the reviewer could kindly reconsider the novelty assessment of our work. Meanwhile, we look forward to discussing other questions if any.
> > >
> > > Sincerely,
> > >
> > > Paper 5388 Authors

---

> > > > ### Comment · Reviewer_bA4X · 2023-08-17
> > > >
> > > > Thank the author for the further explanation. I partly agree with the technical part rather than all parts. Although i still support my viewpoint that it's similar to OTOV2, but i decided to increase my score to 4.

---

> > > > > ### Author Response · Authors · 2023-08-18
> > > > > **Thank you for raising the score. We provide more clarifications.**
> > > > >
> > > > > Dear Reviewer bA4X,
> > > > >
> > > > > We greatly appreciate that you increased the rating of our work and recognizes the practical and technical novelty of OTOv3 compared with the existing NAS methods. For your remaining concern regarding the similarity to OTOv2, we provide the following clarification and hope could adequately address it.
> > > > >
> > > > > We believe that we are on the same page that OTOv3 and OTOv2 tackle two orthogonal challenges of AutoML: OTOv2 for automated structured pruning, and OTOv3 for automated sub-network architecture search. Therefore, their algorithmic designs are natural to be different to cater the divergent scenarios. If so, **the perceived similarity may largely come from their paradigms with similar terminologies**. For instance, both utilize (distinct) dependency graphs, group variables into ZIGs or GeZIGs, and employ sparse optimizers (DHSPG or H2SPG). Despite these similar terminologies in both frameworks, their foundational meanings, designs, and outcomes diverge substantially for distinct goals, which have been detailed in other discussion threads. Please let us know if additional clarification is needed.
> > > > >
> > > > > In the below, we would like to reveal that the similar naming convention is a reflection of our intention and long-term vision to foster a universal framework. While we hadn't previously highlighted this, considering it as future work, we elaborate on it now.
> > > > >
> > > > > **The consistent naming of terminologies in OTOv3 was a deliberate decision**. This choice stems from our long-term vision of fostering a universal framework. **This framework aspires to seamlessly integrate both automated structured pruning and sub-network architecture search for general DNNs**, which are isolately addressed by their respective communities. While these challenges are distinct and orthogonal, they both fall under the overarching theme of identifying the optimal sub-network from a general DNN, but along different dimensions.
> > > > >
> > > > > To offer an analogy based on the computational graph of DNNs: structured pruning essentially 'trims' the vertices (operators), whereas sub-network architecture search eliminates certain vertices (operators) and edges (the connections between operators). We believe that there is an avenue for synergy between them. Our objective is to pave the way to stimulate such a comprehensive, automated framework that bridges both tasks to benefit the AI community.
> > > > >
> > > > > **For making the potential universal framework concise and elegant**, we deliberately unify the components in OTOv3 to be partially aligned with OTOv2. This integration leads to analogous terminologies, even though each module functions distinctly. Unified terminologies present clear potential advantages for the future. Specifically, this allows for the formulation of the universal automated framework in a following sophisticated manner. Otherwise, it would be messed up in our imagination.
> > > > >
> > > > > - **Dependency graph analysis.** Create two distinct dependency graphs for trimming or eliminating operators and connections, respectively.
> > > > > - **Sparse optimization.** Formulate a (single or multi-level) hierarchical overlapping structured sparsity optimization problem and solve it. The overlapping refers that both tasks require distinct variable partitions wherein groups are overlapped. (This is another open and challenging optimization problem, especially for DNN applications.)
> > > > > - **Sub-network construction.** Remove redundant operators and connections and trim partially redundant operators.
> > > > >
> > > > >
> > > > > As a summary of the above, the partially aligned terminologies in OTOv2 and OTOv3 are intentionally designed to pave the way for a future universal framework. Yet, their individual modules vary significantly. **On the journey towards this universal framework, it would also unveil numerous open problems that are valued for study within the community.** Meanwhile, we will further polish the writing to better distinguish the difference of both frameworks and describes their similarities.
> > > > >
> > > > > We sincerely hope the above could adequately address the concern regarding the similarity to OTOv2 and look forward to further discussion.
> > > > >
> > > > > Sincerely,
> > > > >
> > > > > Paper 5388 Authors.

---

> > > > > > ### Comment · Reviewer_bA4X · 2023-08-18
> > > > > >
> > > > > > Thanks for further explanation, and I have no other question. But I insist on my view.

---

> > > > > > > ### Author Response · Authors · 2023-08-18
> > > > > > > **Thank you**
> > > > > > >
> > > > > > > We greatly appreciate your valued efforts and the recognition of our work. We respect your opinion, yet we also believe in the high practical and technical novelty and potential benefits of OTOv3 to the AI community. If any new question or concern is raised, we are always here to answer. Thanks for your efforts again!

---

### Official Review · Reviewer_h4H8 · 2023-07-06

**Soundness:** 1 poor
**Presentation:** 1 poor
**Contribution:** 2 fair
**Rating:** 4
**Confidence:** 3

**Summary:**

This paper proposes OTOv3, an approach to train general supernets and discover promising subnetworks. It claims to be able to automatically generate the search space, and construct subnetworks based on hierarchical half-space projected gradient. The proposed approach has been evaluated on a number of datasets, showing comparable performance to the state of the art.

**Strengths:**

+ The proposed approach has been evaluated on a number of popular datasets.
+ The objectives make sense, and the overall idea of automatically building a search space and then search for strong performing models is interesting.

**Weaknesses:**

- It seems not clear how this 3rd version of OTO advances the previous versions. It seems the delta in this paper is quite minor.
- There are many places throughout the paper, are either not clear or perhaps not correct. For instance, the very first sentence in the abstract:`Existing neural architecture search (NAS) methods typically rely on pre-specified super deep neural networks (super-networks)...` I think in NAS context this can be very misleading. Also the dependency graph construction seems to be quite standard in graph analysis.
- Some of the experimental results reported are not very clear. How the proposed approach could be 4-5x faster than training-free NAS, if compared in a fair way? Does the 0.1 GPU day search cost include everything you need to do to get the final architecture?
- Missing comparison with existing work such as TE-NAS, ZiCo etc.

**Questions:**

See above.

---

> ### Author Rebuttal · Authors · 2023-08-03
>
> Dear Reviewer h4H8,
>
> We appreciate your constructive comments and valued suggestions and responded in details below. Look forward to further discussion.
>
> - **The delta between OTOv3 and OTOv2 are minor**.
>
> > Thank for the question. Please refer to our global rebuttal response along with a PDF which illustrate the difference between OTOv3 and OTOv2. We hope that we adequately addressed your concern.
>
> - **There are many places throughout the paper, are either not clear or perhaps not correct. For example, Existing neural architecture search (NAS) methods typically rely on pre-specified super deep neural networks (super-networks) with handcrafted search-space beforehand.**
>
> > Thanks for the valued suggestion. To the best of our understanding, in order to search an optimal sub-network given a general DNN, existing Neural Architecture Search (NAS) methods typically necessitate the initial handcrafting of a search space. This typically involves significant human intervention. To overcome these challenges, we present a novel algorithmic framework accompanied by an end-to-end automated system called OTOv3. Designed for general DNNs, this system has three main capabilities: (i) automatic construction of the search space, (ii) identification of removal redundant structures while maintaining high-performance training, and (iii) automatic generation of sub-networks in a one-shot manner.
>
> >To better deliver the context, we will rephrase the first sentence as **"To search optimal sub-networks given general DNNs, existing neural architecture search methods typically rely on handcrafting the search spaces beforehand, which usually requires significant human intervention"**.  Meanwhile, we are aware that there exist other NAS methods that construct or grow network candidates given a pre-specified search pool and policy. This setting is different from ours which starts from a general super DNN and ends in an optimal sub-network. We will carefully revisit the manuscript and polish the writing to make the description clearer and more precise.
>
>
> - **How the proposed approach could be 4-5x faster than training-free NAS?**
>
> > Thanks for the insightful question. Please refer to our global rebuttal response above that hopefully could adequately address your concern. Furthermore, the search cost of OTOv3 takes into account the time spent from the construction of the dependency graph to the identification of redundant structures, once the architecture of the sub-network has been determined.
>
> - **Missing comparison with existing work such as TE-NAS, ZiCo etc.**
>
> > Thanks for providing the great literatures. We have provided preliminary comparison with TE-NAS in the global rebuttal response and will include both into the revision.
>
> Sincerely,
>
> Paper 5388 Authors

---

> > ### Comment · Reviewer_h4H8 · 2023-08-18
> > **Thanks for the response**
> >
> > Thanks for the response and additional results. I also checked the comments and discussion from the other reviewers and I still feel that the delta over OTOv2 is limited. Having that said based on the authors response I would increase my score to 4.

---

> > > ### Author Response · Authors · 2023-08-18
> > > **Thank you**
> > >
> > > Dear Reviewer h4H8,
> > >
> > > We greatly appreciate your feedback and are pleased to address most of your concerns. We are also grateful for the improved score. While we respect your perspective on the delta to OTOv2, we believe in the high practical and technical innovations of OTOv3. Meanwhile, we are optimistic about its potential benefits to the AI community. To ensure clarity, we will enhance our manuscript to more effectively highlight the differences between the two approaches, especially in the context of orthogonal target problems, in our revision.
> > >
> > > We will remain available to address any further questions or concerns that arise. Thank you for the valued efforts!
> > >
> > > Sincerely,
> > >
> > > Paper 5388 Authors

---

### Official Review · Reviewer_csCP · 2023-07-09

**Soundness:** 3 good
**Presentation:** 2 fair
**Contribution:** 3 good
**Rating:** 6
**Confidence:** 3

**Summary:**

The authors process a training method to efficiently and automatically find optimal subnetworks without the need to configure the search space or a pre-specified supernetwork.

**Strengths:**

- A clever way to build the search space without manual intervention using zero-invariant group partition
- Avoiding the need to choose the supernetwork beforehand
- Automatically finding the optimal subnetwork
- Good results to show the validity of the method

**Weaknesses:**

- While OTOv3 is able to outperform prior art on smaller datasets like FashionMNIST, CIFAR, SVHN, it doesn't beat them in a larger dataset like ImageNet (It's still competitive). Examples: P-DARTS, AmoebaNet-C, PC-DARTS - similar FLOPS/PARAMs but better than OTOv3
- In practice, accuracy (or relevant metric) is one of the improvement metrics to optimize against. Therefore, would it not be better to have a suite of subnetworks that trade-off accuracy/flops vs just being given one automatically?

**Questions:**

- Can this approach be extended to transformer-based architecture search?
- How can this approach be modified to give a set of subnetworks which can cater to different user needs versus just one subnetwork?

**Limitations:**

- Limitations of the approach have not been adequately discussed

---

> ### Author Rebuttal · Authors · 2023-08-03
>
> Dear Reviewer csCP,
>
> We appreciate your insightful comments and constructive suggestions. Please see our responses to the comments. Look forward to further discussion.
>
> - **Can this approach be extended to transformer-based architecture search?**
>
> > That is a great question. Certainly, OTOv3 can be extended to transformer-based architecture search. In fact, we are actively employing OTO onto Large Language Models (LLMs) and have recorded promising experimental progress.
>
> > With greater details, the majority of operators in the transformer-based architecture are naturally supported, e.g., the MLP layers. The multi-head attention layers require further considerations to form them as an entirety since they are composed by assembling multiple basic operators in the DNN trace graph.
>
> > Furthermore, OTO framework is also flexible to be integrated with other cutting-edge techniques that are widely used to fine-tune large-scale transformers such as [LoRA]. Transformers with LoRA will affect the presence of dependency graph to introduce *overlapping* node groups, which are currently disjoint in the dependency graphs of the current libraries (see the attached rebuttal PDF). We will detail these applications in our forthcoming work.
>
> > [LoRA] LoRA: Low-Rank Adaptation of Large Language Models, ICLR 2021.
>
> - **How can this approach be modified to give a set of subnetworks which can cater to different user needs versus just one subnetwork?**
>
> > That is a great question. The framework is flexible to produce multiple sub-networks upon varying criteria for meeting the user needs. In particular, after automatically constructing the search space given general DNNs, we could modify the redundancy score calculation or integrate with other proxies in accordance with the specific user demands.
>
> - **Some prior works achieve higher accuracy on ImageNet than OTOv3.**
>
> > Thanks for the great question. We have perspective that their outperformance is mainly driven by the usage of multi-level optimization that trains additional pre-specified auxiliary architecture variables to avoid overfitting over training dataset. For the sake of autonomy and generality, the current OTOv3 has though automatically created search space given general DNNs, not automatically introduced architecture variables yet, (which is another open problem). Then we formulated a single-level hierarchical structured-sparsity problem upon the automated search space.
>
> > That being said, OTO framework is flexible to be integrated with multi-level optimization if equips with an automatic introduction of architecture variables. Such an enhancement could potentially lead to further accuracy improvements and is left as a future work upon the current library.
>
> - **Limitations have not been adequately discussed.**
>
> > Thanks for the great question. Besides the limitations described in Appendix A. 1, we will discuss them more adequately and provide potential solutions in the revision. The main limitations are largely from an engineering perspective, including unsupported operators, the requirement for removal structures to contain trainable variables, and potential overfitting to the training dataset. The later two issues could be largely resolved if we could realize automated introduction of architecture variables to enable multi-level optimization in the future.
>
> Sincerely,
>
> Paper 5388 Authors.

---

> > ### Comment · Reviewer_csCP · 2023-08-17
> > **Response to Rebuttal**
> >
> > Thank you authors for the response to my questions. Based on this, I still stand by my original appraisal.

---

> > > ### Author Response · Authors · 2023-08-17
> > >
> > > We thank the reviewer for standing by the original appraisal.

---

### Official Review · Reviewer_XuwW · 2023-07-09

**Soundness:** 4 excellent
**Presentation:** 3 good
**Contribution:** 4 excellent
**Rating:** 7
**Confidence:** 4

**Summary:**

The paper "OTOv3: Towards Automatic Sub-Network Search Within General Super Deep Neural Networks" presents a new automated system called Only-Train-Once (OTOv3) for Neural Architecture Search (NAS). Unlike existing NAS methods that often depend on pre-specified super deep neural networks with handcrafted search spaces, OTOv3 can train general super-networks and generate high-performing sub-networks in a one-shot manner without pre-training and fine-tuning. The authors outline three main contributions of OTOv3: automatic search space construction for general super-networks; a Hierarchical Half-Space Projected Gradient (H2SPG) for ensuring network validity during optimization; and automatic sub-network construction based on the super-network and the H2SPG solution. The effectiveness of OTOv3 is demonstrated on a variety of super-networks and benchmark datasets, with the computed sub-networks achieving competitive or superior performance.

**Strengths:**

1. The proposed method is backed by substantial theoretical considerations and empirical validation, demonstrating the quality of the research.
2. The paper is well-structured and clearly written, making the proposed method and its benefits understandable.
3. OTOv3 can be applied to a wide range of super-networks and has shown competitive or superior performance on several benchmark datasets, indicating its potential impact in the field of NAS.

**Weaknesses:**

1. The term search space and supernet are not well defined in the paper. To my understanding, supernet is a kind of representation of search space. In the paper, the author says that "OTOv3 automatically generates a search space given a general super-network", which is confusing.
2. The authors have not discussed the potential limitations of OTOv3, such as its possible limitations in search space, or potential for overfitting. These factors could impact its practical applicability.

**Questions:**

See weakness.

**Limitations:**

See weakness.

---

> ### Author Rebuttal · Authors · 2023-08-03
>
> Dear Reviewer XuwW,
>
> We appreciate your valued comments and favorable recommendations for our work. Please see our responses to the constructive suggestions. Look forward to further discussion.
>
> - **The term search space and supernet are not well defined in the paper. To my understanding, supernet is a kind of representation of search space.**
>
> > Thanks for the great suggestion. We will define the terminologies better in the revision and unify *super-network* as *general DNN* to avoid ambiguity.
>
> - **The authors have not discussed the potential limitations of OTOv3, such as its possible limitations in search space, or potential for overfitting. These factors could impact its practical applicability.**
>
> > Thanks for the great suggestion. You are right that OTOv3 has limitations in terms of search space and potential overfitting. As detailed in Appendix A.1, the primary constraints of the existing library stem largely from an engineering perspective. Issues such as unsupported operators, the requirement for removal structures to contain trainable variables, and no automated architecture variable introduction may arise. The last one further forced us to formulate a single-level hierarchical structured sparsity optimization problem, which might overfit the training dataset.
>
> > Despite these limitations, we are optimistic and believe that the library would become more and more mature by harnessing contributions from both our end and the wider open-source community, building upon its current state.
>
> Sincerely,
>
> Paper 5388 Authors

---

### Author Rebuttal · Authors · 2023-08-03

Dear reviewers and ACs,

We deeply appreciate all the insightful comments and constructive suggestions that helped us improve our manuscript. We have carefully addressed each comment and will include into the revision. Below, we present our responses to the general questions regarding the difference and novelty of OTOv3 compared with OTOv2 and the search cost against the training-free NAS.

- **Target problems of OTOv3 and OTOv2 are orthogonal.**

>   We would like to kindly emphasize that OTOv3 and OTOv2 address two **distinct and orthogonal** challenges within the field of autoML. We provide the example networks outlined in the attached rebuttal PDF along with further elucidation on this matter below.

> **OTOv2: automated structured pruning for general DNNs.** OTOv2 studied given general DNNs, how to automatically construct a slimmer pruned network. The process of structural pruning eliminates redundancy within each operator while still **maintaining the presence of these operators and the connections between them**.

> **OTOv3: automated sub-network search for general DNNs.** OTOv3 further studies given *general* DNNs, how to *automatically* identify and remove redundant operators entirely to construct a high-performing sub-network. Remark here that in OTOv3, **the operators and connections can be completely removed**. This is a stark contrast to the structured pruning in OTOv2, where they are preserved.

- **What is the novelty compared with OTOv2?**

> The target problem of OTOv3 is orthogonal to that of OTOv2 and not achieved yet by the existing works to best of our knowledge. We establish a fresh algorithmic framework and develop an end-to-end system from scratch with **three main novel contributions.** These contributions are fundamentally different from those of OTOv2, marking a significant shift in our approach.

> **Automated search space construction via dependency graph analysis.** Initially, OTOv3 automatically constructs dependency graphs for general DNNs. These graphs are used to identify 'removal structures'—structures that can be eliminated without disrupting the normal function of the remaining DNNs. Remark here that OTOv3 and OTOv2 establish distinctly different dependency graph algorithms to cater to their orthogonal objectives. The design, creation, and resulting graphs differ significantly. Please see the rebuttal PDF for more details. Afterwards, OTOv3 constructs a search space based on these removal structures within the dependency graph.

> **H2SPG versus DHSPG.** Subsequently, OTOv3 introduces a novel hierarchical half-space projected gradient method (H2SPG) to identify redundant removal structures within the search space. In contrast to the DHSPG in OTOv2, H2SPG further incorporates the DNN hierarchy when generating sparsity, ensuring that the remaining DNN remains valid. In contrast, the ablation study in Appendix D shows that DHSPG can easily result in invalid sub-networks.

> **Automated subnetwork construction.** Finally, OTOv3 automatically constructs sub-networks by removing redundant structures identified by H2SPG. From the engineering development perspective, OTOv3 presents a greater challenge than OTOv2 because it more aggressively modifies the computational graphs of the DNNs.

> These novel components work together to achieve our goal: given a general DNN, automatically train it from scratch and produce a high-performing sub-network in the one-shot manner.

- **Search cost against training-free NAS**

> **Difference of search space.** We appreciate the reviewers for the insightful question. We would like to clarify that OTOv3 and [ZenNAS] performed over similar yet different ResNet search spaces, leading to variations in search costs. [ZenNAS] populates massive ResNet candidates from a search pool and ranks them by varying zero-shot proxies such as Zen-Score, Synflow, and NASWOT. This process took 0.5 GPU days on a V100 GPU for 1M model (about 0.4 GPU days on A100).  Contrarily, to set up the starting DNN for OTOv3, we independently constructed SuperResNets, as depicted in Figure 7 of Appendix. SuperResNets include the optimal architectures derived from [ZenNAS] and aim to discover the most suitable sub-networks using H2SPG over the automated search space.

> **Comparison with other zero-shot methods.** We appreciate the reviewers for introducing us more literatures of training-free NAS and will include them into the revision. For a preliminary comparison, please see the below table. [TE-NAS] searches more efficiently, even under a 1080Ti GPU. Please note that we have not included [ZiCo], as it did not report CIFAR10 on DARTS, but will compare it with other experiments.

> **Flexibility with zero-shot proxies.** Finally, we would like to highlight that OTOv3 focuses on autonomy and generality but not search cost.  In fact, **OTOv3 is flexible to be integrated with zero-shot proxies for efficient search.** After establishing the search space of general DNNs, we could use training-free schemas to efficiently get sub-networks. This is feasible because **the redundant structure identification mechanism in OTOv3 is modular and is not restricted to gradient-based methods.**

>| Method | Type | Search Space | Acc1 (%) | Params | GPU | Search Cost (GPU days) |
>|--|--|--|--|--|--|--|
>| Zen-Score-1M | Zero-Shot | ResNet Pool | 96.2 | 1M | A100 | 0.4 |
>| Zen-Score-2M | Zero-Shot | ResNet Pool | 97.5 |2M | A100 | 0.5 |
>| **TE-NAS** | Zero-Shot |  DARTS | 97.4 | 3.8M | 1080Ti | **0.05**|
>| ISTA-NAS | Grad | DARTS | 97.5 | 3.3M | A100 | 0.1 |
>| PrDARTS | Grad | DARTS | 97.4 | 3.9M | A100 | 0.2 |
>| OTOv3-2M| Grad | SuperResNet | 97.5 | 2M | A100 | 0.1 |


> [ZenNAS] Zen-NAS: A Zero-Shot NAS for High-Performance Deep Image Recognition, ICCV 2021.
>
> [TE-NAS] Neural Architecture Search on ImageNet in Four GPU Hours, ICLR 2021.
>
> [ZiCo] ZiCo: Zero-shot NAS via Inverse Coefficient of Variation on Gradients, ICLR 2023.


Sincerely,

Paper 5388 Authors

---

### Author Response · Authors · 2023-08-10
**Summary of updates from the authors. Look forward to further discussion.**

Dear reviewers and ACs,

We thank all for your suggestions and questions, and we responded in detail.

-  Specifically, we have added more explanations and comparisons regarding OTOv3 and OTOv2. A PDF is attached to illustrate the discrepancy of both frameworks among dependency graphs, group partitions, sparse optimizers, and produced sub-networks.

- Meanwhile, we clarified the search cost comparison, and will polish the writing to more adequately discuss the limitations and add more descriptive and illustrative languages into the revision.

We would appreciate if reviewers could please take a look, and hopefully our responses have adequately addressed the questions. We thank all again for the valued efforts!

Sincerely,

Paper 5388 Authors

---

### Author Response · Authors · 2023-08-17
**Further clarification regarding novelty perspective of our work.**

Dear reviewers and ACs,

We thank you very much again for your efforts and constructive comments that help us improve the manuscript. We are glad to see that the motivations, potential impacts, and the soundness of algorithmic framework have been well recognized by the most reviewers. As only a few days left in the reviewer-author discussion period, we would like to further clarify the concern regarding the novelty of our work.

We believe the novelty can be largely evaluated from two perspectives: **practical novelty** — determining whether the paper introduces or reveals a previously unachieved scenario or uncovers unknown properties; and **technical novelty** — evaluating if the proposed solution introduces innovative methods to achieve its objective.


**We kindly argue that our work embodies both practical novelty and technical novelty.**


- **Practical novelty.** Prior to OTOv3, one can not easily search a sub-network given a general DNN without handcrafting a search space beforehand. Such inconvenience also prevents the usage of many NAS methods onto wider DNNs and applications. To the best of our knowledge, OTOv3 is the first work that resolves such inconvenience. It can automatically search a high-performing sub-network given a general DNN in the one-shot manner, thereby has high practical novelty.


- **Technical novelty.** To achieve the objective of OTOv3, we establish a fresh end-to-end algorithmic framework with three novel technical contributions compared to the existing NAS works: (i) automatic search space construction via dependancy graph analysis; (ii) H2SPG for solving hierarchical structured sparsity optimization problem; and (iii) automated sub-network construction. Each component was dedicately designed and engineeringly challenging to be implemented.

  We understand that the paradigm might seem a little similar to OTOv2 for automated structured pruning. But since both frameworks tackle distinct and orthogonal target problems, their each individual component has fundamentally different algorithmic designs and implementations, which has made a significant shift to our work. Therefore, we believe that our work has high technical novelty as well.


We hope our clarification could address the concern regarding the novelty of our work. Look forward to further discussion.

Sincerely,

Paper 5388 authors

---

### Decision · Program_Chairs · 2023-09-21

**Decision:**

Reject

**Comment:**

The paper aims to address the limitations of existing NAS methods that rely on pre-specified super deep neural networks with handcrafted search spaces. The paper then proposes OTOv3 that includes automatic search space construction, a Hierarchical Half-Space Projected Gradient that leverages the dependency graph to ensure the network validity during optimization, and automatic sub-network construction based on the super-network and the H2SPG solution. Experiments show the efficacy of the method.

Reviewers are mainly concerned on the difference of the proposed OTOv3 from its previous versions; they are also concerned on the less clear definitions of search space and super network, and the less competitive performance on larger dataset such as ImageNet. The authors give thorough responses that, unfortunately, cannot fully convince the reviewers. AC has the same major concern and is thus not recommending acceptance.